# Parameter Symmetry and Noise Equilibrium of Stochastic Gradient Descent

**Liu Ziyin**
Massachusetts Institute of Technology,
NTT Research
ziyinl@mit.edu

**Mingze Wang**
Peking University
mingzewang@stu.pku.edu.cn

**Hongchao Li**
The University of Tokyo
lhc@cat.phys.s.u-tokyo.ac.jp

**Lei Wu**
Peking University
leiwu@math.pku.edu.cn

## Abstract

Symmetries are prevalent in deep learning and can significantly influence the learning dynamics of neural networks. In this paper, we examine how exponential symmetries – a broad subclass of continuous symmetries present in the model architecture or loss function – interplay with stochastic gradient descent (SGD). We first prove that gradient noise creates a systematic motion (a "Noether flow") of the parameters $\theta$ along the degenerate direction to a unique initialization-independent fixed point $\theta^*$. These points are referred to as the *noise equilibria* because, at these points, noise contributions from different directions are balanced and aligned. Then, we show that the balance and alignment of gradient noise can serve as a novel alternative mechanism for explaining important phenomena such as progressive sharpening/flattening and representation formation within neural networks and have practical implications for understanding techniques like representation normalization and warmup.

## 1 Introduction

Stochastic gradient descent (SGD) and its variants have become the cornerstone algorithms used in deep learning. In the continuous-time limit, the algorithm can be written as [19, 13, 21, 32, 9]:

$$\mathrm{d}\theta_t = -\nabla L(\theta_t)\,\mathrm{d}t + \sqrt{2\sigma^2\Sigma(\theta_t)}\,\mathrm{d}W_t, \tag{1}$$

where $\Sigma(\theta)$ is the covariance matrix of gradient noise (Section 3) with the prefactor $\sigma^2 = \eta/(2S)$ modeling the impact of a finite learning rate $\eta$ and batch size $S$; $W_t$ denotes the Brownian motion. When $\sigma = 0$, Eq. (1) corresponds to gradient descent (GD)[1]. However, SGD and GD can exhibit significantly different behaviors, often converging to solutions with significantly different levels of performance [31, 39, 44, 22, 49]. Notably, even when $\sigma^2 \ll 1$, where we expect a close resemblance between SGD and GD over finite time [19], their long-time behaviors still differ substantially [26]. These observations indicate that gradient noise can bias the dynamics significantly, and revealing its underlying mechanism is thus crucial for understanding the disparities between SGD and GD.

**Contribution.** In this paper, we study the how of SGD noise biases training through the lens of symmetry. Our key contributions are summarized as follows. We show that

1. when symmetry exists in the loss function, the dynamics of SGD can be precisely characterized and is different from GD along the degenerate direction;

38th Conference on Neural Information Processing Systems (NeurIPS 2024).

---

[1]"Gradient descent" and "gradient flow" are used interchangeably as we work in the continuous-time limit.

2. the treatment of common symmetries, including the rescaling and scaling symmetry in deep learning, can be unified in a single theoretical framework that we call the exponential symmetry;
3. for any $\theta$, every exponential symmetry implies the existence of a unique and attractive fixed point along the degenerate direction for SGD;
4. symmetry and balancing of noise can serve as novel mechanisms for important deep learning phenomena such as progressive sharpening/flattening and latent representation formation.

See Figure 1 for an illustration of how symmetry leads to a systematic flow of SGD. This work is organized as follows. We discuss the most relevant works in Section 2. The main theoretical results are presented in Section 4. We apply our theory to understand specific problems and present numerical results in Section 5. The last section concludes this work. All the proofs are presented in the Appendix.

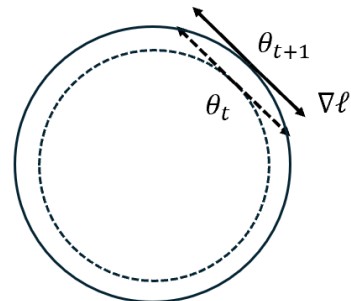

## 2    Related Works

The dynamics of SGD in the degenerate directions of the loss landscape is a poorly understood problem. There are two closely related prior works. Ref. [48] studies the dynamics of SGD when there is a simple rescaling symmetry and applies it to derive the stationary distribution of SGD for linear networks. Our result is more general because rescaling symmetry is the simplest case of exponential symmetries[2]. Another related work is Ref. [20], which studies a different special case of exponential symmetry, the scale invariance, and in the presence of weight decay. Their analysis assumes the existence of the fixed point of the dynamics, which we proved to exist. Also related is the study of conservation laws under gradient flow [34, 18, 24, 43, 40], which we will discuss more closely in Section 4. However, these works do not take the stochasticity of training into account. Comparing with these results that assume no stochasticity, our result could suggest that SGD converges to initialization-independent solutions, whereas the GD finds solutions are strongly initialization-dependent. In addition, Section D extends our main result to discrete-time SGD.

Figure 1: An example of a 2d loss function with scale invariance: $\ell(\theta) = \ell(\lambda\theta)$ for a scalar $\lambda$ and $\theta \in \mathbb{R}^2$. Because of the symmetry, the gradient $\nabla\ell$ must be *tangential* to the circles whose center is the origin. This implies that the norm $\|\theta\|$ does not change during gradient flow training. However, when the training is stochastic or discrete-time, SGD must move outward. If the model starts at $\theta_t$, it must move to a larger circle. As an illustrative example, this loss function has a unique and attractive fixed point: $\|\theta\| = \infty$. SGD will diverge after training under scale invariance. Also, see Remark 4.4 for a discussion of the difference between discrete-time and continuous-time dynamics.

## 3    Preliminaries

**Setup and Notations.**    Let $\ell : \Omega \times \mathcal{Z} \mapsto \mathbb{R}$ denote the per-sample loss, with $\Omega$ and $\mathcal{Z}$ denoting the parameter and sample space, respectively. Here, $z \in \mathcal{Z}$ includes both the input and label and accordingly. We use $\mathbb{E}_z = \mathbb{E}$ to denote the expectation over a given training set. Therefore, $L(\theta) = \mathbb{E}_z[\ell(\theta, z)]$ is the empirical risk function. The covariance of gradient noise is given by

$$\Sigma(\theta) = \mathbb{E}_z[\nabla\ell(\theta, z)\nabla\ell(\theta, z)^\top] - \nabla L(\theta)\nabla L(\theta)^\top.$$

Additionally, we use $\Sigma_v(\theta) := \mathbb{E}_z[\nabla_v\ell(\theta, z)\nabla_v\ell(\theta, z)^\top] - \nabla_v L(\theta)\nabla_v L(\theta)^\top$ to denote the covariance of gradient noise impacting on the subset of parameters $v$. Denote by $(\theta_t)_{t \geq 0}$ the trajectory of SGD or GD. For any $h : \Omega \mapsto \mathbb{R}$, we write $h_t = h(\theta_t)$ and $\dot{h}(\theta_t) = \frac{\mathrm{d}}{\mathrm{d}t}h(\theta_t)$ for brevity. When the context is clear, we also use $\ell(\theta)$ to denote $\ell(\theta, z)$.

**Symmetry.**    The per-sample loss $\ell(\cdot, \cdot)$ is said to possess the $Q$-symmetry if

$$\ell(\theta, z) = \ell(Q_\rho(\theta), z), \forall\rho \in \mathbb{R}, \tag{2}$$

where $(Q_\rho)_{\rho \in \mathbb{R}}$ is a set of continuous transformation parameterized by $\rho \in \mathbb{R}$. Without loss of generality, we assume $Q_0 = \mathrm{id}$. The most common symmetries exist within the model $f$, namely $f_\theta$ is invariant under certain transformations of $\theta$. However, our formalism is slightly more general in the sense that it is also possible for the model to be variant while the per-sample loss remains unchanged, which appears in self-supervised learning [50], for example.

---

[2]These are known as "continuous symmetries." Prior works also studied SGD training under discrete symmetries [45, 3], which are different from continuous symmetries.

# 4 Continuous Symmetry and Noise Equilibria

Taking the derivative with respect to $\rho$ at $\rho = 0$ in Eq. (2), we have

$$0 = \nabla_\theta \ell(\theta, z) \cdot J(\theta), \tag{3}$$

where $J(\theta) = \frac{\mathrm{d} Q_\rho(\theta)}{\mathrm{d}\rho}|_{\rho=0}$. Denote by $C$ be the antiderivative of $J$, that is, $\nabla C(\theta) = J(\theta)$. Then, taking the expectation over $z$ in (3) gives the following conservation law for GD solutions $(\theta_t)_{t\geq0}$:

$$\dot{C}(\theta_t) = 0. \tag{4}$$

Essentially, this is a consequence of Noether's theorem [25], and $C$ will be called a "Noether charge" in analogy to theoretical physics. The conservation law (4) implies that the GD trajectory is constrained on the manifold $\{\theta : C(\theta) = C(\theta_0)\}$. We refer to Ref. [18] for a study of this type of conservation law under the Bregman Lagrangian [16].

## 4.1 Noether Flow in Degenerate Directions

In this paper, we are interested in how $C(\theta_t)$ changes, if it changes at all, under SGD. By Ito's lemma, we have the following *Noether flow* (namely, the flow of the Noether charge):

$$\dot{C}(\theta_t) = \sigma^2 \mathrm{Tr}\left[\Sigma(\theta_t)\nabla^2 C(\theta_t)\right], \tag{5}$$

where $\nabla^2 C$ denotes the Hessian matrix of $C$. The derivation is deferred to Appendix B. By definition, $\Sigma(\theta_t)$ is always positive semidefinite (PSD). Thus, we immediately have: if $\nabla^2 C$ is PSD throughout training, $C(\theta_t)$ is a monotonically increasing function of time. Conversely, if $\nabla^2_\theta C$ is negative semidefinite (NPD), $C(\theta_t)$ is a monotonically decreasing function of time.

The existence of symmetry implies that (with suitable conditions of smoothness) any solution $\theta$ resides within a connected, loss-invariant manifold, defined as $\mathcal{M}_\theta := \{Q_\rho(\theta) : \rho \in \mathbb{R}\}$. We term directions within this manifold as "degenerate directions" since movement along them does not change the loss value. Notably, the biased flow (5) suggests that SGD noise can drive SGD to explore within this manifold along these degenerate directions since the value of $C(\theta)$ for $\theta \in \mathcal{M}_\theta$ can vary.

## 4.2 Exponential symmetries

Now, let us focus on a family of symmetries that is common in deep learning. Since the corresponding conserved quantities are quadratic functions of the model parameters, we will refer to this class of symmetries as *exponential symmetries*.

**Definition 4.1.** $(Q_\rho)_\rho$ is said to be a exponential symmetry if $J(\theta) := \frac{\mathrm{d}}{\mathrm{d}\rho}Q_\rho(\theta)|_{\rho=0} = A\theta$ for a symmetric matrix $A$.

This implies when $\rho \ll 1$, $Q_\rho = \mathrm{id} + \rho A + o(\rho)$. In the sequel, we also use the words "$A$-symmetry" and "$Q$-symmetry" interchangeably since all properties of $Q_\rho$ we need can be derived from $A$. This definition applies to the following symmetries that are common in deep learning:

- *Rescaling symmetry*: $Q_\rho(a, b) = (a(\rho+1), b/(\rho+1))$, which appears in linear and ReLU networks [7, 48]. In this symmetry, $A = \mathrm{diag}(I_a, -I_b)$, where $I$ is the identity matrix with dimensions matching that of $a$ and $b$.
- *Scaling symmetry*: $Q_\rho\theta = (\rho + 1)\theta$, which exists whenever part of the model normalized using techniques like batch normalization [14], layer normalization [2], or weight normalization [29]. In this case, $A = I$.
- *Double rotation symmetry:* This symmetry appears when parts of the model involve a matrix factorization problem, where for an arbitrary invertible matrix $B$ $\ell = \ell(UW) = \ell(UBB^{-1}W)$. Writing the exponential symmetry for this case is a little cumbersome. We need first to view $U$ and $W$ as a single vector, and the exponential transformation is given by a block-wise diagonal matrix $\mathrm{diag}(B, ..., B, B^{-1}, ..., B^{-1})$. See Section 5.1 for more detail.

It is possible for only a subset of parameters to have a given symmetry. Mathematically, this corresponds to the case when $A$ is low-rank. It is also common for $\ell$ to have multiple exponential symmetries at once, often for different (but not necessarily disjoint) subsets of parameters. For example, a ReLU network has a different rescaling symmetry for every hidden neuron.

It is obvious that under this $Q$ symmetry, the Noether charge has a simple quadratic form:

$$C(\theta) = \theta^\top A\theta. \tag{6}$$

Moreover, the interplay between this symmetry and weight decay can be explicitly characterized in our framework. To this end, we need the following definition.

**Definition 4.2.** For any $\gamma \in \mathbb{R}$, we say $\ell_\gamma(\theta, x) := \ell(\theta, x) + \gamma\|\theta\|^2$ has the $Q$ symmetry as long as $\ell(\theta, x)$ has the $Q$ symmetry.

For the SGD dynamics that minimizes $L_\gamma(\theta) = \mathbb{E}_x[\ell_\gamma(\theta, x)]$, it follows from (5) that

$$\dot{C}(\theta_t) = -4\gamma C(\theta_t) + \sigma^2 \mathrm{Tr}[\Sigma(\theta_t)A] =: G(\theta_t). \tag{7}$$

Thus, a positive $\gamma$ always causes $|C(\theta_t)|$ to decay, and the influence of symmetry is determined by the spectrum of $A$. Denote by $A = \sum_j \mu_j n_j n_j^\top$ the eigendecomposition of $A$. Then,

$$\mathrm{Tr}[\Sigma(\theta_t)A] = \sum_{i:\mu_i>0} \mu_i n_i^\top \Sigma(\theta_t)n_i + \sum_{j:\mu_j<0} \mu_j n_j^\top \Sigma(\theta_t)n_j.$$

This gives a clear interpretation of the interplay between SGD noise and the exponential symmetry: the noise along the positive directions of $A$ causes $C(\theta_t)$ to grow, while the noise along the negative directions causes $C(\theta_t)$ to decay. In other words, the noise-induced dynamics of $C(\theta_t)$ is determined by the competition between the noise along the positive- and negative-eigenvalue directions of $A$.

**Time Scales.** The above analysis implies that the dynamics of SGD can be decomposed into two parts: the dynamics that directly reduce loss, and the dynamics along the degenerate direction of the loss, which is governed by Eq (5). These two dynamics have essentially independent time scales. The first part is independent of the $\sigma^2$ in expectation, whereas the time scale of the dynamics in the degenerate directions depends linearly on $\sigma^2$.

The first time scale $t_\mathrm{erm}$ is due to the dynamics of empirical risk minimization. The second time scale $t_\mathrm{equi}$ is the time scale for Eq. (5) to reach equilibrium, which is irrelevant to direct risk minimization. When the parameters are properly tuned, $t_\mathrm{erm}$ is of order 1, whereas $t_\mathrm{equi}$ is proportional to $\sigma^2 = \eta/(2S)$. Therefore, when $\sigma^2$ is large, the parameters will stay close to the equilibrium point early in the training, and one can expect that $\dot{C}(\theta_t)$ is approximately zero after $t_\mathrm{equi}$. In line with Ref. [20], this can be called the fast-equilibrium phase of learning. Likewise, when $\sigma^2 \ll 1$, the approach to equilibrium will be slower than the actual time scale of risk minimization, and the dynamics in the degenerate direction only take off when the model has reached a local minimum. This can be called the slow-equilibrium phase of learning.

### 4.3 Noise Equilibrium and Fixed Point Theorem

It is important and practically relevant to study the stationary points of dynamics in Eq. (7). Formally, the stationary point is reached when $-\gamma C(\theta) + \eta \mathrm{Tr}[\Sigma(\theta)A] = 0$. Because we make essentially no assumption about $\ell(\theta)$ and $\Sigma(\theta)$, one might feel that it is impossible to guarantee the existence of a fixed point. Remarkably, we prove below that a fixed point exists and is unique for every connected degenerate manifold.

To start, consider the exponential maps generated by $A$:

$$e^{\lambda A}\theta := \lim_{\rho \to 0}(I + \rho A + o(\rho))^{\lambda/\rho}\theta,$$

which applies the symmetry transformation to $\theta$ for $\lambda/\rho$ times. Then, it follows that if we apply $Q_\rho$ transformation to $\theta$ infinitely many times and for a perturbatively small $\rho$,

$$\ell(\theta) = \ell\left(e^{\lambda A}\theta\right). \tag{8}$$

Thus, the exponential symmetry implies the symmetry with respect to an exponential map, a fundamental element of Lie groups [11]. Note that exponential-map symmetry is also an exponential symmetry by definition. For the exponential map, the degenerate direction is clear: for any $\lambda$, $\theta$ connects to $e^{\lambda A}\theta$ without any loss function barrier. Therefore, the degenerate direction for any exponential symmetry is unbounded. Now, we prove the following fixed point theorem, which shows that for every exponential symmetry and every $\theta$, there is one and only one corresponding fixed point in the degenerate direction.

**Theorem 4.3.** *Let the per-sample loss satisfy the A-exponential symmetry and $\theta_\lambda := \exp[\lambda A]\theta$. Then, for any $\theta$ and any $\gamma \geq 0$,[3]*

*(1) $G(\theta_\lambda)$ (Eq. (7)) and $-C(\theta_\lambda)$ are monotonically decreasing functions of $\lambda$;*
*(2) there exists a $\lambda^* \in \mathbb{R} \cup \{\pm\infty\}$ such that $G(\theta_{\lambda^*}) = 0$;*
*(3) in addition, if $G(\theta_\lambda) \neq 0$, $\lambda^*$ is unique and $G(\theta_\lambda)$ is strictly monotonic;*
*(4) in addition to (3), if $\Sigma(\theta)$ is differentiable, $\lambda^*(\theta)$ is a differentiable function of $\theta$.*

*Remark* 4.4. It is now worthwhile to differentiate gradient flow (GF), GD, SGD, and stochastic gradient flow (SGF). Technically, one can prove that the same result holds for discrete-time GD and SGD in expectation, and GF is the only of the four algorithms that do not obey this theorem (See Section D), and so one could argue that the discrete step size is the essential cause of noise balance. Mathematically, the SGF can be seen as a model of the leading order effect of having a finite step size and thus also share this effect (remember that the Ito Lemma contains a second-order term in $d\theta$). That being said, there is a practical caveat: in practice, we find it much easier for models to reach these fixed points with SGD than with GD, and so it is fair to say that this effect is the most dominant when gradient noise is present.

Part (1), together with Part (2), implies that the unique stationary point is essentially attractive. This is because $\dot{C}$ decreases with $\lambda$ while $C$ increases with it. Let $C^* = C(\theta_{\lambda^*})$. Thus, $C(\theta) - C^*$ always have the opposite sign of $\lambda^*$, while $\frac{d}{dt}C(\theta)$ will have the same sign. Conceptually, this means that $C$ will always move to reduce its distance to $C^*$. Assuming that $C^*$ is a constant in time (or close to a constant, which is often the case at the end of training), Part (1) implies that $\frac{d}{dt}(C(\theta) - C^*) \propto -\mathrm{sgn}(C(\theta) - C^*)$, signaling a convergence to $C(\theta) = C^*$. In other words, SGD will move to restore the balance if it is perturbed away from $\lambda^* = 0$. If the matrix $\Sigma A$ is well-behaved, one can indeed establish the convergence to the fixed point in the relative distance even if $C^*$ is mildly divergent due to diffusion. Because this part is strongly technical and our focus is on the fixed points, we leave the formal statement and its discussion to Appendix B.3.

**Theorem 4.5.** *(Informal) Let $C^*$ follow a drifted Brownian motion and $\Sigma A$ satisfy two well-behaved conditions. Then, either $C - C^* \to 0$ in $L_2$ or $(C - C^*)^2/(C^*)^2 \to 0$ in probability.*

Parts (2) and (3) show that a unique fixed point exists. We note that it is more common than not for the conditions of uniqueness to hold because there is generally no reason for $\mathrm{Tr}[\Sigma(\theta)A]$ or $\mathrm{Tr}[\theta\theta^\top A]$ to vanish simultaneously, except in some very restrictive subspaces. One major (perhaps the only) reason for the first trace to vanish is when the model is located at an interpolation minimum. However, interpolation minima are irrelevant for modern large-scale problems such as large language models because the amount of available text for training far exceeds the size of the largest models. Even when the interpolation minimum exists, the unique fixed point should still exist when the training is not complete. See Figure 1. Part (4) means that the fixed points of the dynamics is well-behaved. If the parameter $\theta$ has a small fluctuation around a given location, $C$ will also have a small fluctuation around the fixed point solution. This justifies approximating $C$ by a constant value when $\theta$ changes slowly and with small fluctuation.

**Fixed point as a Noise Equilibrium.** Let $\theta^*$ be a fixed point of (7). It must satisfy

$$4\gamma C(\theta^*) = \sigma^2 \mathrm{Tr}\left[\Sigma(\theta^*)A\right]. \tag{9}$$

Hence, a large weight decay leads to a small $|C(\theta^*)|$, whereas a large gradient noise leads to a large $|C(\theta^*)|$. When there is no weight decay, we get a different equilibrium condition: $\mathrm{Tr}[\Sigma(\theta^*)A] = 0$, which can be finite only when $A$ contains both positive and negative eigenvalues. This equilibrium condition is equivalent to $\sum_{i:\mu_i>0} \mu_i n_i^\top \Sigma(\theta^*)n_i = -\sum_{j:\mu_j<0} \mu_j n_j^\top \Sigma(\theta^*)n_j$. Namely, the overall gradient fluctuation in the two different subspaces specified by the symmetry $A$ must balance. We will see that the main implication of this result is that the gradient noise between different layers of a deep neural network should be balanced at the end of training. Conceptually, Theorem 4.3 suggests the existence of a special type of fixed point for SGD, which the following definition formalizes.

**Definition 4.6.** $\theta$ is a *noise equilibrium* for a nonconstant function $C(\theta)$ if $\dot{C}(\theta) = 0$ under SGD.

## 5 Applications

Now, we analyze the noise equilibria of a few important problems. These examples are prototypes of what appears frequently in deep learning practice and substantiate our arguments with numerical

---

[3]A similar result can be proved for the discrete-time SGD. See Section D.

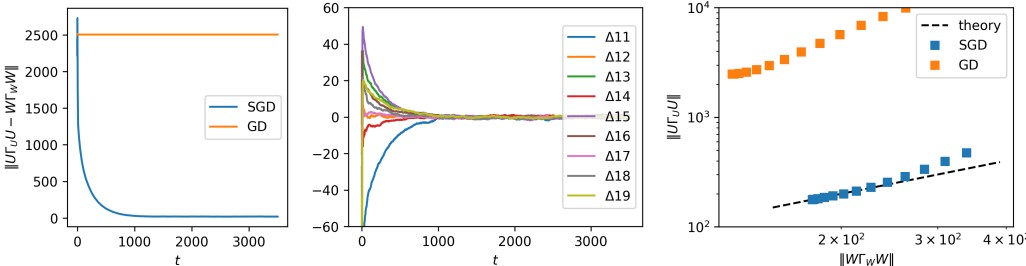

Figure 2: Comparison between GD and SGD for matrix factorizations. **Left**: Example of a learning trajectory. The convergence speed is almost exponential-like in experiments. **Mid**: evolution of 10 individual elements of $\Delta_{ij} \coloneqq (U^\top \Gamma_U U - W \Gamma_W W^\top)_{ij}$. As the theory shows, they all move close to zero and fluctuate with a small variance. **Right**: Converged solutions of SGD agree with the prediction of Theorem 5.2, but are an order of magnitude away from the solution found by GD, even if they start from the same init.

examples. In addition, an experiment with the scale invariance in normalized tanh networks is presented in Appendix A.1.

## 5.1 Generalized Matrix Factorization

Exponential symmetry is also observed when the model involves a (generalized) matrix factorization. This occurs in standard matrix completion problems [33] or within the self-attention of transformers through the key and query matrices [35]. For a (generalized) matrix factorization problem, we have the following symmetry in the objective:

$$\ell(U, W, \theta') = \ell(UA, A^{-1}W, \theta') \tag{10}$$

for any invertible matrix $A$ and symmetry-irrelevant parameters $\theta'$. We consider matrices $A$ that are close to identity: $A = I + \rho B + O(\rho^2)$, and $A^{-1} = I - \rho B + O(\rho^2)$. Therefore, for an arbitrary symmetric $B$, we have a conserved quantity for GD: $C_B(\theta) = \text{Tr}[UBU^\top] - \text{Tr}[W^\top BW]$. This conservation law can also be written in the matrix form, which is a well-known result for GD [8, 24]:

$$(W_t W_t^\top - U_t^\top U_t) = (W_0 W_0^\top - U_0^\top U_0). \tag{11}$$

For SGD, applying (5) gives the following proposition.

**Proposition 5.1.** *Suppose the symmetry* (10) *holds. Let* $U = (\tilde{u}_1, \cdots, \tilde{u}_{d_2})^\top \in \mathbb{R}^{d_2 \times d}$, $W = (\tilde{w}_1, \cdots, \tilde{w}_{d_0}) \in \mathbb{R}^{d \times d_0}$, *where* $\tilde{u}_i, \tilde{w}_j \in \mathbb{R}^d$. *Let* $C_B(\theta) = \text{Tr}[UBU^\top] - \text{Tr}[W^\top BW]$ *for any symmetric matrix* $B \in \mathbb{R}^{d \times d}$. *Then, for SGD, we have*

$$\dot{C}_B(\theta_t) = \sigma^2 \left( \sum_{i=1}^{d_2} \text{Tr}[\Sigma_{\tilde{u}_i}(\theta_t)B] - \sum_{j=1}^{d_0} \text{Tr}[\Sigma_{\tilde{w}_j}(\theta_t)B] \right).$$

This dynamics is analytically solvable when $U \in \mathbb{R}^{1 \times d}$ and $W \in \mathbb{R}^{d \times 1}$. In this case, taking $B = E_{k,l} + E_{l,k}$ where $E_{i,j}$ denotes the matrix with entries of 1 at $(i,j)$ and zeros elsewhere. For this choice of $B$, we obtain that $C_B(\theta) = W_k W_l - U_k U_l$, and applying the results we have derived, it is easy to show that for some random variable $r$: $\dot{C}_B(\theta_t) = -\text{Var}[r(\theta_t)]C_B(\theta_t)$, which signals an exponential decay. For common problems, $\text{Var}[r(\theta_t)] > 0$ [48]. Since the choice of $B$ is arbitrary, we have that $W_k W_l \to U_k U_l$ for all $k$ and $l$. The message is rather striking: SGD automatically converges to a solution where all neurons output the same sign $(\text{sgn}(U_i) = \text{sgn}(U_j))$ at an exponential rate.

## 5.2 Balance and Stability of Matrix Factorization

As a concrete example, let us consider a two-layer linear network (this can also be seen as a variant of standard matrix factorizations):

$$\ell_\gamma = \|UWx - y\|^2 + \gamma(\|U\|_F^2 + \|W\|_F^2). \tag{12}$$

where $x \in \mathbb{R}^{d_x}$ is the input data, and $y = y' + \epsilon \in \mathbb{R}^{d_y}$ is a noisy version of the label. The ground truth mapping is linear and realizable: $y' = U^* W^* x$. The second moments of the input and noise are denoted as $\Sigma_x = \mathbb{E}[xx^\top]$ and $\Sigma_\epsilon = \mathbb{E}[\epsilon\epsilon^\top]$, respectively. Note that this problem is essentially identical to a matrix factorization problem, which is not only a theoretical model of neural networks but also an important algorithm frequently in use for recommender systems [41]. The following theorem gives the fixed point of Noether flow.

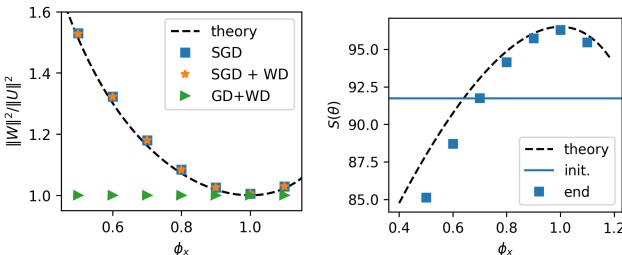

Figure 3: A two-layer linear network after training. Here, the problem setting is the same as Figure 8. The theoretical prediction is computed from Theorem 5.2. **Left**: balance of the norm is only achieved when $\phi_x = 1$, namely, when the data has an isotropic covariance. We also test SGD with a small weight decay ($10^{-4}$), which is sufficiently small that the solution we obtained for SGD without SGD still holds approximately. In contrast, training with GD + WD always converges to a norm-balanced solution. **Right**: the sharpness of the converged model trained with SGD. We see that for some data distributions, SGD converges to a sharper solution, whereas it converges to flatter solutions for other data distributions. This flattening and sharpening effect are both due to the noise-balance effect of SGD. Here, we find that the systematic error between experiment and theory is due to the use of a finite learning rate and decreases as we decrease $\eta$.

**Theorem 5.2.** *Let $r = UWx - y$ be the prediction residual. For all symmetric $B$, $\dot{C}_B = 0$ if*

$$W\Gamma_W W^\top = U^\top \Gamma_U U, \tag{13}$$

*where $\Gamma_W = \mathbb{E}[\|r\|^2 xx^\top] + 2\gamma I$, $\Gamma_U = \mathbb{E}[\|x\|^2 rr^\top] + 2\gamma I$.*

See Figure 8 for the convergence of SGD to this solution under different learning rate, batch size and width. The equilibrium condition takes a more suggestive form when the model is at the global minimum, where $U^*W^*x - y = \epsilon$. Assuming that $\epsilon$ and $x$ are independent and that there is no weight decay, we have:

$$W\bar{\Sigma}_x W^\top = U^\top \bar{\Sigma}_\epsilon U \tag{14}$$

Here, the bar over the matrices indicates that they have been normalized by their traces: $\bar{\Sigma} = \Sigma/\text{Tr}[\Sigma]$. The matrices $\Gamma_W$ and $\Gamma_U$ simplifies because at the global minimum, $r_i = \epsilon_i$ and so $\mathbb{E}[\|x\|^2 rr^\top] = \text{Tr}[\Sigma_x]\Sigma_\epsilon$ and $\mathbb{E}[\|r\|^2 xx^\top] = \text{Tr}[\Sigma_\epsilon]\Sigma_x$. This condition should be compared with the alignment condition for GD in Eq. (11), where the alignment is entirely determined by the initialization and perfect alignment is achieved only if the initialization is perfectly aligned. This condition simplifies further if both $\bar{\Sigma}_x$ and $\bar{\Sigma}_\epsilon$ are isotropic, where the equation simplifies to $WW^\top/d_x = U^\top U/d_y$. Namely, the two layers will be perfectly aligned, and the overall balance depends only on input and output dimensions. Figure 2-Left shows an experiment that shows that the two-layer linear net is perfectly aligned after training. Here, every point corresponds to the converged solution of an independent run with the same initialization and training procedures but different values of $\Sigma_\epsilon$. In agreement with the theory, the two layers are aligned according to Theorem 5.2 under SGD, but not under GD. In fact, GD finds solutions that are more than an order of magnitude away from SGD.

**Noise Driven Progressive Sharpening and Flattening.** This result implies a previously unknown mechanism of progressive sharpening and flattening, where, during training, the stability of the algorithm steadily improves (during flattening) or deteriorates (during sharpening) [39, 15, 6]. To see this, we first derive a metric of sharpness for this model.

**Proposition 5.3.** *For the per-sample loss* (12)*, let $S(\theta) := \text{Tr}[\nabla^2 L(\theta)]$. Then, $S(\theta) = d_y\|W\Sigma_x^{1/2}\|_F^2 + \|U\|_F^2 \text{Tr}[\Sigma_x]$.*

The trace of the Hessian is a good metric of the local stability of the GD and SGD algorithm because the trace upper bounds the largest Hessian eigenvalue. Let us analyze the simplest case of an autoencoding task, where the model is at the global minimum. Here, $\Sigma_x \propto I_{d_x}$, $\Sigma_\epsilon \propto I_{d_y}$. For a random Gaussian initialization with variance $\sigma_W^2$ and $\sigma_U^2$, the trace at initialization is, in expectation, $S_{\text{init}} = d_y d\text{Tr}[\Sigma_x](\sigma_W^2 + \sigma_U^2)$. At the end of the training, the model is close to the global minimum and satisfies Proposition 5.3. Here, the rank of $U$ and $W$ matters and is upper bounded by $\min(d, d_x)$, and at the global minimum, $U$ and $W$ are full-rank (equal to $\min(d, d_x)$), and all the singular values are 1. Thus,

$$\begin{cases} S_{\text{init}} = d_x d(\sigma_U^2 + \sigma_W^2)\text{Tr}[\Sigma_x], \\ S_{\text{end}} = 2\min(d, d_x)\text{Tr}[\Sigma_x]. \end{cases} \tag{15}$$

The change in the sharpness during training thus depends crucially on the initialization scheme. For Xavier init, $\sigma_U^2 = (d_y + d)^{-1}$ and $\sigma_W^2 = (d + d_x)^{-1}$, and so $S_{\text{init}} \approx S_{\text{end}}$ (but $S_{\text{init}}$ is slightly smaller). Thus, for the Xavier init., the sharpness of loss experiences a small sharpening during training. For Kaiming init., $\sigma_U^2 = 1$ and $\sigma_W^2 = d_x^{-1}$. Therefore, it always holds that $S_{\text{init}} \geq S_{\text{end}}$, and so the stability improves as the training proceeds. The only case when the Kaiming init. does not experience progressive flattening is when $d = d_x = d_y$, which agrees with the common observation that training is easier if the widths of the model are balanced [12]. See Figure 4 for an experiment. In previous works, the progressive sharpening happens when the model is trained with GD [6]; our theory suggests an alternative mechanism for it.

A practical technique that the theory explains is using warmup to stabilize training in the early stage. This technique was first proposed in Ref. [10] for training CNNs, where it was observed that the training is divergent if we start the training at a fixed large learning rate $\eta_{\max}$. However, this divergent behavior disappears if we perform a warmup training, where the learning rate is increased gradually from a minimal value to $\eta_{\max}$. Later, the same technique is found to be crucially useful for training large language models [27]. Our theory shows that the gradient noise can drive Kaiming init. to a stabler status where a larger learning can be applied.

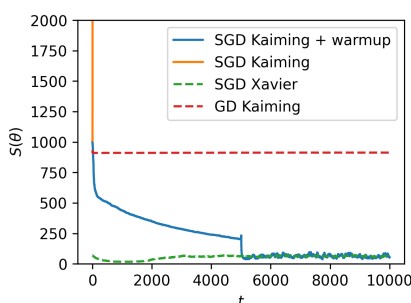

Figure 4: Dynamics of the stability condition $S$ during the training of a rank-1 matrix factorization problem. The solid lines show the training of SGD with Kaiming init. When the learning rate ($\eta = 0.008$) is too large, SGD diverges (orange line). However, when one starts training at a small learning rate (0.001) and increases $\eta$ to 0.008 after 5000 iterations, the training remains stable. This is because SGD training improves the stability condition during training, which is in agreement with the theory. In contrast, the stability condition of GD and that of SGD with a Xavier init increases only slightly. Also, note that both Xavier and Kaiming init. under SGD converges to the same stability condition because the equilibrium is unique.

**Flat or Sharp?** Prior works have often argued that SGD prefers flatter solutions to sharper ones (e.g., see Ref. [42]). The exact solution we found, however, implies a subtle picture: for some datasets, SGD prefers sharper solutions, while for others, SGD prefers flatter solutions. Therefore, there is no causal relationship between SGD training and the sharpness of the found solution. See Figure 3 for the dependence of the flatness on the data distribution. A related question is whether SGD noise creates a similar effect as weight decay training. The answer is also negative: weight decay always prefers smaller norms and, thus, norm-balanced solutions, which are not necessarily noise-aligned solutions. Figure 3 shows that SGD can also lead to unbalanced solutions, unlike weight decay.

## 5.3 Noise-Aligned Solution of Deep Linear Networks

Here, we apply our result to derive the exact solution of an arbitrarily deep and wide deep linear network, which has been under extensive study due to its connection in loss landscape to deep neural networks [4, 5, 17, 23, 47, 38]. Deep linear networks have also been a major model for understanding the implicit bias of GD [1]. The per-sample loss for a deep linear network can be written as:

$$\ell(\theta) = \|W_D...W_1 x - y\|^2, \tag{16}$$

where $W_i$ is an arbitrary dimensional matrix for all $i$. The global minimum is realizable: $y = Vx + \epsilon$, for i.i.d. noises $\epsilon$. Because there is a double rotation symmetry between every two neighboring matrices, the Noether charge can be defined with respect to every such pair of matrices. Let $B_i$ be a symmetric matrix; we define the charges to be $C_{B_i} = W_i^T B_i W_i$. The noise equilibrium solution is given by the following theorem.

**Theorem 5.4.** *Let $W_D...W_1 = V$. Let $V' = \sqrt{\Sigma_\epsilon} V \sqrt{\Sigma_x}$ such that $V' = LS'R$ is its SVD and $d = \text{rank}(V')$. Then, for all $i$ and all $B_i$, a noise equilibrium for $C_{B_i}$ at the global minimum is*

$$\sqrt{\Sigma_\epsilon} W_D = L\Sigma_D U_{D-1}^\top, \ W_i = U_i \Sigma_i U_{i-1}^\top, \ W_1 \sqrt{\Sigma_x} = U_1 \Sigma_1 R, \tag{17}$$

*for $i = 2, \cdots, D-1$. $U_i$ are arbitrary matrices satisfying $U_i^T U_i = I_{d \times d}$, and $\Sigma_i$ are diagonal matrices such that*

$$\Sigma_1 = \Sigma_D = \left(\frac{d}{\text{Tr}S'}\right)^{(D-2)/2D} \sqrt{S'}, \ \Sigma_i = \left(\frac{\text{Tr}S'}{d}\right)^{1/D} I_{d \times d}. \tag{18}$$

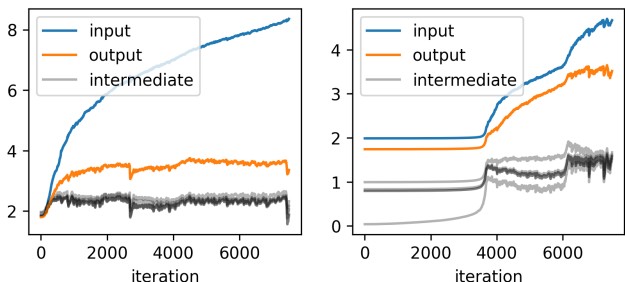

Figure 5: Norms of weights of multilayer deep linear network during training on MNIST without weight decay. We see that the intermediate layers converge to the same norm during training, whereas the input and output layers are different because they are determined by the input and output noise. This effect is robust against different initializations. This agrees with our analysis for deep linear nets (**Theorem** 5.4). **Left**: initializing all layers with the same norm. **Right**: initializing all layers at randomly different norms.

This solution has quite a few striking features. Surprisingly, the norms of all intermediate layers are balanced:

$$\mathrm{Tr}[\Sigma_1^2] = \mathrm{Tr}[\Sigma_i^2] = (\mathrm{Tr}S')^{2/D}d^{1-2/D}. \tag{19}$$

All intermediate layers are thus rescaled orthogonal matrices aligned with the neighboring matrices and the only two matrices that process information are the first and the last layer. See Figure 5 for an illustration of this effect. This explains an experimental result first observed in Ref. [30], where the authors showed that the neural networks find similar solutions when the model is initialized with the standard init., where there is no alignment at the start, and with the aligned init. Thus, the balance and alignment between different layers in the neural networks can be attributed to the rescaling symmetry between each pair of matrices.

## 5.4 Approximate Symmetry and Bias of SGD

Lastly, let us consider what happens if the loss function only has an approximate symmetry. As a minimal model, let us consider the following loss function: $\ell = \ell_1(\theta, x) + \zeta\ell_2(\theta)$. Here, $\ell_1$ has the $A$-symmetry, whereas $\ell_2(\theta)$ has no symmetry nor randomness and so $\ell_2$ does not affect $\Sigma$ at all. $\zeta$ determines the relative strength between the two terms. In totality, $\ell$ no longer has the $A$-symmetry.

As before, let $C_A = \theta^\top A\theta$. Then, $\dot{C}_A(\theta) = -\zeta(\nabla\ell_2)^\top A\theta^* + \sigma^2\mathrm{Tr}[\Sigma(\theta)A]$, whose fixed point is

$$\zeta(\nabla\ell_2)^\top A\theta^* = \sigma^2\mathrm{Tr}[\Sigma(\theta)A]. \tag{20}$$

This equilibrium condition thus depends strongly on how large $\zeta$ is. When $\zeta$ is small, we see that SGD still favors the fixed point given by Theorem 4.3, but with a first-order correction in $\zeta$.

Conversely, if $\zeta$ is large and $\sigma^2$ is small, we can expand around a local minimum of the loss function $\theta^*$, and so the fixed point becomes

$$\zeta(\theta - \theta^*)^\top H(\theta^*)A\theta^* = \sigma^2\mathrm{Tr}[\Sigma(\theta^*)A] + O(\sigma^2\|\theta - \theta^*\| + \|\theta - \theta^*\|^2), \tag{21}$$

where $H$ is the Hessian of $\ell_2$. Certainly, this implies that SGD will stay around a point that deviates from the local minimum by an $O(\sigma^2)$ amount. This stationary point potentially has many solutions. For example, one class of solution is when $\theta - \theta^*$ is an eigenvector of $H$ with eigenvalue $h^* > 0$ and eigenvector $n$, we can denote $s = (\theta - \theta^*)^\top n$ and obtain a direct solution of $s$:

$$s = \frac{\sigma^2\mathrm{Tr}[\Sigma(\theta^*)A]}{\zeta h^* n^\top A\theta^*}. \tag{22}$$

This deviation disappears in the limit $\sigma^2 \to 0$. Therefore, this implicit regularization effect is only a consequence of SGD training and is not present under GD. With this condition, one can obtain a clear expression of the deviation of the quantity $C$ from its local minimum value $C^* := (\theta^*)^\top A\theta^*$. We have that

$$C(\theta) = C^* + 2(\theta - \theta^*)^\top A\theta^* + O(\|\theta - \theta^*\|^2) = C^* + 2\frac{\sigma^2}{\zeta h^*}\mathrm{Tr}[\Sigma A]. \tag{23}$$

Thus, our results in the previous section still apply. The quantity $C$ will be systematically larger than the local minimum values of $C$ if the approximate symmetry matrix $A$ is PD. It is systematically smaller if $A$ is ND. When $A$ contains both positive and negative eigenvalues, the deviation of $C$

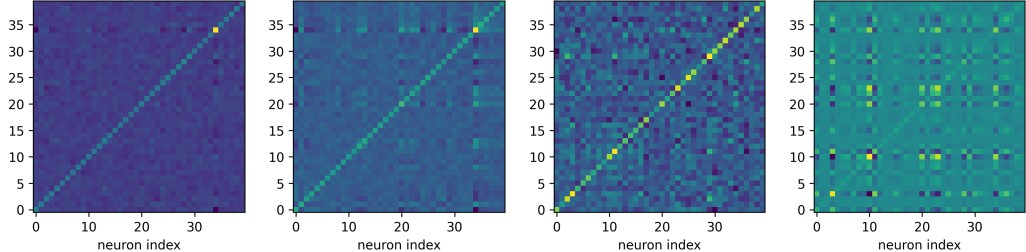

Figure 6: The latent representations of a two-layer tanh net trained under SGD (**left**) are similar across different layers, in agreement with the theory. However, the learned representations are dissimilar under GD (**right**). Here, we plot the matrices $W\bar{\Sigma}_x W$ (first and third plots) and $\bar{U}\bar{\Sigma}_\epsilon U$ (second and fourth plots). Note that the quantity $W\bar{\Sigma}_x W$ is equal to the covariance of the preactivation representation of the first layer. This means that SGD and GD learn qualitatively different features after training. Also, see Appendix A.4 for other activations. This mechanism also complements the recent result in Ref. [46], which proposes a physics-inspired theory showing that gradient noise is a key factor in determining the latent representation of neural networks.

depends on the local gradient fluctuation balancing condition. When the smallest eigenvalue of $H$ is close to zero (which is true for common neural networks), the dominant factor that biases $C$ occurs in this space. Therefore, it is not bad to approximate the deviation as $C(\theta) \approx C^* + 2\sigma^2 \mathrm{Tr}[\Sigma A]/h_{\min}$, where $h_{\min}$ is the smallest eigenvalue of the Hessian at the local minimum. In reality, $\zeta$ is neither too large nor too small, and one expects that the solution favored by SGD is an effective interpolation between the true local minimum and the fixed point favored by the symmetries.

A set of experiments is shown in Figure 6, where we compare the latent representation of a two-layer tanh net with the prediction of 5.2. This is a natural example because fully connected networks are believed to be approximated by deep linear networks because they have the same connectivity patterns. We thus compare the prediction of Theorem 5.2 with the experimental results of nonlinear networks. Here, the task is a simple autoencoding task, where $x \in \mathbb{R}^{40}$ and $y = x + \epsilon$. $x$ is sampled from an isotropic Gaussian, and $\epsilon$ is an independent non-isotropic (but diagonal) Gaussian noise such that $\mathrm{Var}[\epsilon_1] = 5$ and $\mathrm{Var}[\epsilon_i] = 1$ for $i \neq 1$. We train with SGD or GD for $10^4$ iterations. The experimental results show that if trained with SGD, the learned representation agrees with the prediction of Theorem 5.2 well, whereas under GD, the model learned a completely different representation. This suggests that our result may be greatly useful for understanding the structures of latent representations of trained neural networks because the quantity $W\bar{\Sigma}_x W$ has a clean interpretation as the normalized covariance matrix of pre-activation hidden representation. Also, this result is not a special feature of tanh networks. Appendix A.4 also shows that the same phenomenon can be observed for swish [28], ReLU, and leaky-ReLU nets.

# 6 Conclusion

In this work, we have studied how continuous symmetries affect the learning dynamics and fixed points of SGD. The result implies that SGD converges to initialization-independent solutions at the end of training, in sharp contrast to GD, which converges to strongly initialization-dependent solutions. We constructed the theoretical framework of exponential symmetries to study the special tendency of SGD to stay close to a special fixed point along the constant directions of the loss landscape. We proved that every exponential symmetry leads to a mapping of every parameter to a unique and essentially attractive fixed point. This point also has a clean interpretation: it is the point where the gradient noises of SGD in different subspaces *balance* and *align*. Because of this property, we termed these fixed points the "noise equilibria." The advantage of our result is that it only relies on the existence of symmetries and is independent of the particular definitions of model architecture or data distribution. A limitation of our work is that we only focus on the problems that exponential symmetries can describe. It would be important to extend the result to other types of symmetries in the future. Another interesting future direction is to study these noise equilibria of more advanced models, which may deepen both our understanding of deep learning and neuroscience.

## Acknowledgement

Lei Wu is supported by the National Key R&D Program of China (No. 2022YFA1008200) and National Natural Science Foundation of China (No. 2288101). Hongchao Li is supported by Forefront Physics and Mathematics Program to Drive Transformation (FoPM), a World-leading Innovative Graduate Study (WINGS) Program, the University of Tokyo. Mingze Wang is supported in part by

the National Key Basic Research Program of China (No. 2015CB856000). We thank anonymous reviewers for their valuable comments.

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

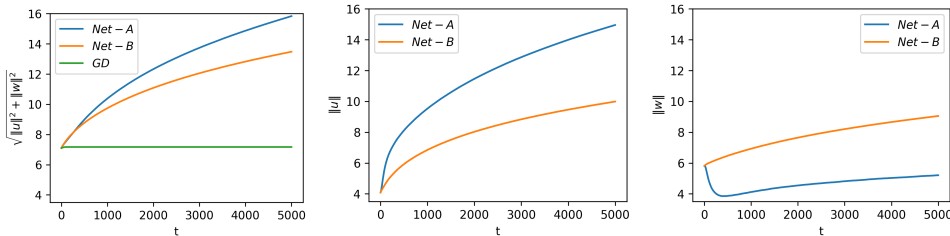

Figure 7: When there is scaling symmetry, the norm of the parameters increases monotonically under SGD but remains unchanged under GD. **Left**: evolution of the total model norm for two-layer nonlinear networks where there is a rescaling symmetry. **Mid**: evolution of the second layer. **Right**: evolution of the first layer. This shows that the evolution of each layer can be vastly different, but the total norm of the parameters with the scaling symmetry is always monotonically increasing. Also, note that for net $B$, each layer also has the rescaling symmetry, and so the norm of each layer for net-$B$ is also increasing. In contrast, net-$A$ does not have layer-wise symmetry, and the individual norms can be either increasing or decreasing.

## A    Additional Experiments

### A.1    Scale Invariance

The scale invariance appears when common normalization techniques such as batch normalization [14] and layer normalization [2] are used. Let $\ell(\theta, x)$ denote a per-sample loss such that for any $\rho \in \mathbb{R}^+$: $\ell(\theta, x) = \ell(\rho\theta, x)$, where $\theta \in \mathbb{R}^d$. For this symmetry, $A = I$. Thus, by Eq. (7), we have during SGD training that

$$\frac{\mathrm{d}}{\mathrm{d}t}\|\theta_t\|^2 = -\gamma\|\theta_t\|^2 + \sigma^2 \mathrm{Tr}[\Sigma(\theta_t)]. \tag{24}$$

Thus, without weight decay, the parameter norm increases monotonically and even diverges, particularly for under-parameterized models where the gradient noise is typically non-degenerate.

Here, we numerically compare two networks trained on GD and SGD: Net-A: $f(x) = \sum_j \frac{u_j}{\|w\|} \tanh(w_j^\intercal x/\|u\|_F)$; and Net-B: $f(x) = \sum_j \frac{u_{ij}}{\|u\|} \tanh(w_j^\intercal x/\|w\|_F)$. Here, $w$ and $u$ are matrices and $w_j$ denotes the $j$-th row of $w$ and $u_j$ denotes the $j$-th column of $u$. The two networks are different functions of $u$ and $w$. However, both networks have the global scale invariance: if we scale both $U$ and $W$ by an arbitrary positive scalar $\rho$, the network output and loss function remain unchanged for any sample $x$. We train these two networks on simple linear Gaussian data with GD or SGD. Figure 7 shows the result. Clearly, for SGD, both networks have a monotonically increasing norm, whereas the norm remains unchanged when the training proceeds with GD. What's more, Net-B has two additional layer-wise scale invariances where one can scale only $u$ (or only $w$) by $\rho$ while keeping the loss function unchanged. This means that both layers will have a monotonically increasing norm, which is not the case for Net-A.

Recent works have studied the dynamics of SGD under the scale-invariant models when weight decay is present [36, 20]. Our result shows that the model parameters will diverge without weight decay, leading to potential numerical problems. Combining the two results, the importance of having weight decay becomes clear: it prevents the divergence of models.

### A.2    Experiment Detail for Alignment Dynamics of Matrix Factorization

Here, we give the details for the experiment in Figure 2. We train a two-layer linear net with $d_0 = d_2 = 30$ and $d = 40$. The input data is $x \sim \mathcal{N}(0, 1)$, and $y = x + \epsilon$, where $\epsilon$ is i.i.d. Gaussian with unit variance. At the end of SGD training, every element of the matrix $U^\intercal \Gamma_U U$ is close to that of $W\Gamma_W W^\intercal$, and they are therefore very well aligned. Such a phenomenon does not happen for GD.

### A.3    Convergence of Matrix Factorization to the Noise Equilibrium

See Figure 8.

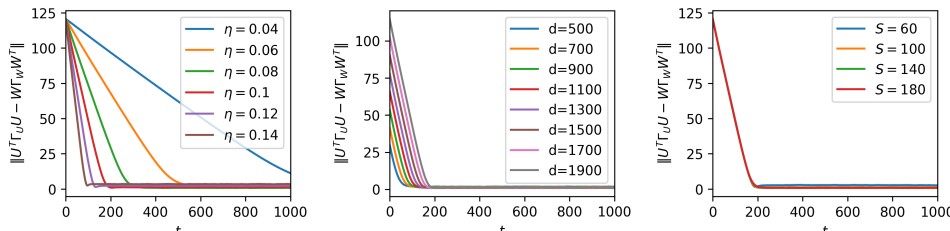

Figure 8: The convergence of matrix factorization to the noise equilibria is robust against different hyperparameter settings. The task is an autoencoding task where $y = x \in \mathbb{R}^{100}$. The distribution of $x$ is controlled by a parameter $\phi_x$: $x_{1:50} \sim \mathcal{N}(0, \phi_x)$, $x_{51:100} \sim \mathcal{N}(0, 2 - \phi_x)$. This directly controls the overall covariance of $x$. The output noise covariance is set to be identity. Unless it is the independent variable, $\eta$, $S$ and $d$ are set to be 0.1, 100 and 2000, respectively. **Left**: using different learning rates. **Mid**: different data dimension: $d_x = d_y = d$. **Right**: different batch size $S$.

## A.4   Alignment in Nonlinear Networks

Here, we complement the experiment in the main text with other types of activations. The experimental setting is exactly the same except that we switch the activation to swish, ReLU, and Leaky-ReLU. This shows that the prediction of Proposition 13 may have a surprisingly wide applicability. See Figure 9.

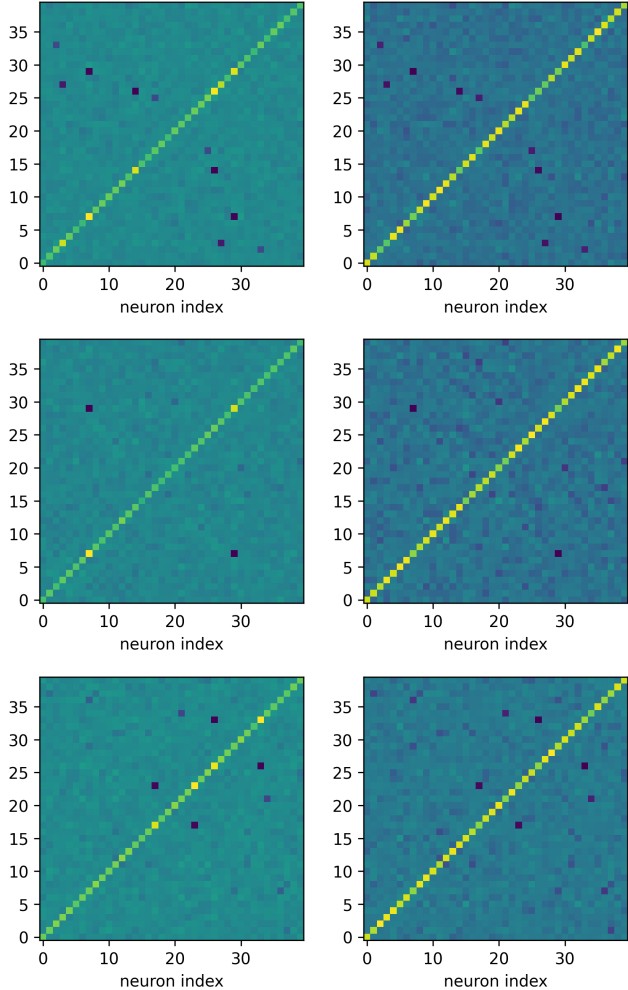

Figure 9: Activation patterns of nonlinear networks trained with SGD (**Upper to lower**: ReLU, leaky-ReLU, swish). **Left**: $W\Gamma_W W^\top$. **Right**: $U^\top \Gamma_U U$. The similarity between the two matrices is striking.

# B Proofs

## B.1 Ito's Lemma and Derivation of Eq. (5)

Let a vector $X_t$ follow the following stochastic process:

$$\mathrm{d}X_t = \mu_t \, \mathrm{d}t + G_t \, \mathrm{d}W_t \tag{25}$$

for a matrix $G_t$. Then, the dynamics of any function of $X_t$ can be written as (Ito's Lemma)

$$\mathrm{d}f(X_t) = \left( \nabla_X^\top f \mu_t + \frac{1}{2} \mathrm{Tr}[G_t^\top \nabla^2 f(X_t) G_t] \right) \mathrm{d}t + \nabla f(X_t)^\top G_t \, \mathrm{d}W_t. \tag{26}$$

Applying this result to quantity $C(\theta)$ under the SGD dynamics, we obtain that

$$dC = \left( \nabla^\top C \nabla L + \frac{\sigma^2}{2} \mathrm{Tr}[\Sigma(\theta)\nabla^2 C] \right) dt + \nabla^\top C \sqrt{\sigma^2 \Sigma(\theta)} dW_t, \tag{27}$$

where we have used $\mu_t = \nabla L$, $G_t = \sqrt{\sigma^2 \Sigma(\theta)}$. By Eq. (3), we have that

$$\nabla^\top C \nabla L = \mathbb{E}[\nabla^\top C \nabla \ell] = 0, \tag{28}$$

and

$$\nabla^\top C \Sigma = \mathbb{E}[\nabla^\top C \nabla \ell \nabla^\top \ell] - \mathbb{E}[\nabla^\top C \nabla \ell] \mathbb{E}[\nabla^\top \ell] = 0. \tag{29}$$

Because $\Sigma(\theta)$ and $\sqrt{\Sigma(\theta)}$ share eigenvectors, we have that

$$\nabla^\top C \sqrt{\sigma^2 \Sigma(\theta)} = 0. \tag{30}$$

Therefore, we have derived:

$$dC = \frac{\sigma^2}{2} \mathrm{Tr}[\Sigma(\theta)\nabla^2 C] dt. \tag{31}$$

## B.2 Proof of Theorem 4.3

We first prove a lemma that links the gradient covariance at $\theta$ to the gradient covariance at $\theta_\lambda$.

**Lemma B.1.**

$$\mathrm{Tr}[\Sigma(\theta_\lambda)A] = \mathrm{Tr}[e^{-2\lambda A}\Sigma(\theta)A]. \tag{32}$$

*Proof.* By the definition of the exponential symmetry, we have that for an arbitrary $\lambda$,

$$\ell(\theta) = \ell(e^{\lambda A}\theta). \tag{33}$$

Taking the derivative of both sides, we obtain that

$$\nabla_\theta \ell(\theta) = e^{\lambda A} \nabla_{\theta_\lambda} \ell(\theta_\lambda), \tag{34}$$

The standard result of Lie groups shows that $e^{\lambda A}$ is full-rank and symmetric, and its inverse is $e^{-\lambda A}$. Therefore, we have

$$e^{-\lambda A} \nabla_\theta \ell(\theta) = \nabla_{\theta_\lambda} \ell(\theta_\lambda). \tag{35}$$

Now, we apply this relation to the trace of interest. By definition,

$$\Sigma(\theta_\lambda) = \mathbb{E}[\nabla_{\theta_\lambda} \ell(\theta_\lambda) \nabla_{\theta_\lambda}^\top \ell(\theta_\lambda)] \tag{36}$$

$$= e^{-\lambda A} \Sigma(\theta) e^{-\lambda A}. \tag{37}$$

Because $e^{\lambda A}$ is a function of $A$, it commutes with $A$. Therefore,

$$\mathrm{Tr}[\Sigma(\theta_\lambda)A] = \mathrm{Tr}[e^{-\lambda A}\Sigma(\theta)e^{-\lambda A}A] \tag{38}$$

$$= \mathrm{Tr}[e^{-2\lambda A}\Sigma(\theta)A]. \tag{39}$$

$\square$

Now, we are ready to prove the main theorem.

*Proof.* First of all, it is easy to see that $C(\theta_\lambda)$ is a monotonically increasing function of $\lambda$. By definition,

$$C(\theta_\lambda) = \theta^T e^{\lambda A} A e^{\lambda A} \theta \tag{40}$$

$$= \theta^T (Z_+ + Z_-)\theta, \tag{41}$$

where we have decomposed the matrix $e^{\lambda A} A e^{\lambda A} = Z_+ + Z_-$ into two symmetric matrices such that $Z_+$ only contains nonnegative eigenvalues, and $Z_-$ only contains nonpositive eigenvalues. Because $e^{\lambda A}$ commute with $A$, they share the eigenvectors. Using elementary Lie algebra shows that the eigenvalues of $Z_+$ are $a_+ e^{\lambda a_+}$ and that of $Z_-$ are $a_- e^{\lambda a_-}$, where $a_+ \geq 0$ and $a_0 \leq 0$. This implies that $\theta^T Z_+ \theta$ and $\theta^T Z_- \theta$ are monotonically increasing functions of $\lambda$.

Now, by Lemma B.1, we have

$$\mathrm{Tr}[\Sigma(\theta_\lambda)A] = \mathrm{Tr}[e^{-2\lambda A}\Sigma(\theta)A]. \tag{42}$$

Similarly, the regularization term is

$$\gamma \theta_\lambda^\top A \theta_\lambda = \gamma \mathrm{Tr}[\theta\theta^\top A e^{2\lambda A}]. \tag{43}$$

Now, by assumption, if $G(\theta_\lambda) \neq 0$, we have either $\mathrm{Tr}[\Sigma(\theta)A] \neq 0$ or $\mathrm{Tr}[\theta\theta^\top A] \neq 0$.

If $\mathrm{Tr}[\Sigma(\theta)A] = \theta^\top A\theta = 0$, we have already proved item (2) of the theorem. Therefore, let us consider the case when either (or both) $\mathrm{Tr}[\Sigma(\theta)A] \neq 0$ or $\theta^\top A\theta \neq 0$

Without loss of generality, we assume $\gamma \geq 0$, and the case of $\gamma < 0$ follows an analogous proof. In such a case, we can write the trace in terms of the eigenvectors $n_i$ of $A$:

$$-\gamma\theta_\lambda^\top A\theta_\lambda + \eta\mathrm{Tr}[\Sigma(\theta_\lambda)A] = \underbrace{\eta \sum_{\mu_i>0} e^{-2\lambda|\mu_i|}|\mu_i|\sigma_i^2 + \gamma \sum_{\mu_i<0} e^{-2\lambda|\mu_i|}|\mu_i|\tilde{\theta}_i^2}_{I_1(\lambda)} - \underbrace{\left(\eta \sum_{\mu_i<0} e^{2\lambda|\mu_i|}|\mu_i|\sigma_i^2 + \gamma \sum_{\mu_i>0} e^{2\lambda|\mu_i|}|\mu_i|\tilde{\theta}_i^2\right)}_{I_2(\lambda)}$$

$$=: I(\lambda),$$

where $\mu_i$ is the $i$-th eigenvalue of $A$, $\tilde{\theta}_i = (n_i^\top \theta_i)^2$, $\sigma_i^2 = n_i^\top \Sigma(\theta)n_i \geq 0$ is the norm of the projection of $\Sigma$ in this direction.

By definition, $I_1$ is either a zero function or strictly monotonically increasing function with $I_1(-\infty) = +\infty$, $I_1(+\infty) = 0$ Likewise, $I_2$ is either a zero function or a strictly monotonically increasing function with $I_2(-\infty) = 0$, $I_2(+\infty) = +\infty$. By the assumption $\mathrm{Tr}(\Sigma(\theta)A) \neq 0$ or $\mathrm{Tr}(\theta\theta^\top A) \neq 0$, we have that at least one of $I_1$ and $I_2$ must be a strictly monotonic function.

- If $I_1$ or $I_2$ is zero, we can take $\lambda$ to be either $+\infty$ or $-\infty$ to satisfy the condition.

- If both $I_1$ and $I_2$ are nonzero, then $I = I_1 - I_2$ is a strictly monotonically decreasing function with $I(-\infty) = +\infty$ and $I(+\infty) = -\infty$. Therefore, there must exist only a unique $\lambda^* \in \mathbb{R}$ such that $I(\lambda^*) = 0$.

For the proof of (4), we denote the multi-variable function $J(\theta; \lambda) := G(\theta_\lambda)$. Given that $\Sigma(\theta)$ is differentiable, $\frac{\partial J}{\partial \theta}$ exists.

It is easy to see that $\frac{\partial J}{\partial \lambda}$ is continuous. Moreover, for any $\theta$ and $\lambda = \lambda^*(\theta)$,

$$-\frac{\partial J}{2\partial \lambda} = \eta \sum_{\mu_i>0} e^{-2\lambda|\mu_i|}|\mu_i|^2\sigma_i^2 + \gamma \sum_{\mu_i<0} e^{-2\lambda|\mu_i|}|\mu_i|^2\tilde{\theta}_i^2 + \eta \sum_{\mu_i<0} e^{2\lambda|\mu_i|}|\mu_i|^2\sigma_i^2 + \gamma \sum_{\mu_i>0} e^{2\lambda|\mu_i|}|\mu_i|^2\tilde{\theta}_i^2 \neq 0.$$

Consequently, according to the Implicit Function Theorem, the function $\lambda^*(\theta)$ is differentiable. Additionally, $\frac{\partial\lambda}{\partial\theta} = -\frac{\frac{\partial J}{\partial\theta}}{\frac{\partial J}{\partial\lambda}}$.

$\square$

### B.3 Convergence

First of all, notice an important property, which follows from Theorem 4.3: $\lambda^* = 0$ if and only if $C - C^* = 0$.

**Lemma B.2.** *For all $\theta(t)$,*

$$\frac{\dot{C}(\theta)}{\lambda^*(\theta)} \geq \begin{cases} 2\sigma^2 \mathrm{Tr}[\Sigma(\theta^*)A_+^2] & \text{if } \lambda^* > 0; \\ 2\sigma^2 \mathrm{Tr}[\Sigma(\theta^*)A_-^2] & \text{if } \lambda^* < 0. \end{cases} \tag{44}$$

*Proof.* As in the main text, let $\theta^*$ denote $\theta_{\lambda^*}$, $C = C(\theta)$ and $C^* = C(\theta^*)$. Thus,

$$\frac{dC}{dt} = \sigma^2 \mathrm{Tr}[\Sigma(\theta)A] \tag{45}$$

$$= \sigma^2 \mathrm{Tr}[\Sigma(\theta^*)e^{2\lambda^* A}A], \tag{46}$$

where the second equality follows from Lemma B.1. One can decompose $A$ as a sum of two symmetric matrices

$$A = \underbrace{Q\Sigma_+ Q^\top}_{:=A_+} + \underbrace{Q\Sigma_- Q^\top}_{:=A_-}, \tag{47}$$

where $Q$ is an orthogonal matrix, $\Sigma_+$ ($\Sigma_-$) is diagonal and contains only non-negative (non-positive) entries. Note that by the definition of $\lambda^*$, we have $\mathrm{Tr}[\Sigma(\theta^*)A] = 0$ and, thus,

$$\mathrm{Tr}[\Sigma(\theta^*)A_+] = -\mathrm{Tr}[\Sigma(\theta^*)A_-]. \tag{48}$$

Thus,

$$\mathrm{Tr}[\Sigma(\theta)A] = \mathrm{Tr}[\Sigma(\theta^*)e^{2\lambda^* A}A] \tag{49}$$

$$= \mathrm{Tr}[\Sigma(\theta^*)(e^{2\lambda^* A} - I)A] \tag{50}$$

$$= \mathrm{Tr}[\Sigma(\theta^*)(e^{2\lambda^* A_+} - I)A_+] + \mathrm{Tr}[\Sigma(\theta^*)(e^{2\lambda^* A_-} - I)A_-]. \tag{51}$$

Using the inequality $I + A \leq e^A$ (namely, that $e^A - I - A$ is PSD), we obtain a lower bound

$$\mathrm{Tr}[\Sigma(\theta)A] \geq 2\lambda^* \mathrm{Tr}[\Sigma(\theta^*)A_+^2] - \mathrm{Tr}[\Sigma(\theta^*)(I - e^{2\lambda^* A_-})A_-] \tag{52}$$

If $\lambda^* > 0$, $\mathrm{Tr}[\Sigma(\theta^*)(e^{2\lambda^* A_-} - I)A_-] < 0$,

$$\mathrm{Tr}[\Sigma(\theta)A] \geq 2\lambda^* \mathrm{Tr}[\Sigma(\theta^*)A_+^2]. \tag{53}$$

Likewise, there is an upper bound, which simplifies to the following form if $\lambda^* < 0$:

$$\mathrm{Tr}[\Sigma(\theta)A] \leq 2\lambda^* \mathrm{Tr}[\Sigma(\theta^*)A_-^2]. \tag{54}$$

This finishes the proof. $\qquad \square$

**Lemma B.3.** *For any $\theta$,*

$$-\frac{C - C^*}{\lambda^*} \leq \begin{cases} 2(\theta^*)^\top A_+^2 \theta^* & \text{if } \lambda^* > 0; \\ 2(\theta^*)^\top A_-^2 \theta^* & \text{if } \lambda^* < 0. \end{cases} \tag{55}$$

*Proof.* The proof is conceptually similar to the previous one. By definition, we have

$$C - C^* = (\theta^*)^\top e^{-2\lambda^* A}A\theta^* - (\theta^*)^\top A\theta^* \tag{56}$$

$$= (\theta^*)^\top A(e^{-2\lambda^* A} - I)\theta^* \tag{57}$$

$$= (\theta^*)^\top A_+(e^{-2\lambda^* A_+} - I)\theta^* + (\theta^*)^\top A_-(e^{-2\lambda^* A_-} - I)\theta^*. \tag{58}$$

By the inequality $I + A \leq e^A$, we have an upper bound

$$C - C^* \geq -2\lambda^*(\theta^*)^\top A_+^2 \theta^* + (\theta^*)^\top A_-(e^{-2\lambda^* A_-} - I)\theta^*. \tag{59}$$

If $\lambda^* > 0$, $(\theta^*)^\top A_-(e^{-2\lambda^* A_-} - I)\theta^* \geq 0$,

$$C - C^* \geq -2\lambda^*(\theta^*)^\top A_+^2 \theta^*. \tag{60}$$

Likewise, if $\lambda^* < 0$, one can prove a lower bound:

$$C - C^* \leq -2\lambda^*(\theta^*)^\top A_-^2 \theta^*. \tag{61}$$

$\qquad \square$

Combining the above two lemmas, one can prove the following corollary.

**Corollary B.4.**

$$\frac{\dot{C}}{C - C^*} \leq \begin{cases} -\sigma^2 \frac{\text{Tr}[\Sigma(\theta^*)A_+^2]}{(\theta^*)^\top A_+^2 \theta^*}, & \text{if } C - C^* < 0; \\ -\sigma^2 \frac{\text{Tr}[\Sigma(\theta^*)A_-^2]}{(\theta^*)^\top A_-^2 \theta^*}, & \text{if } C - C^* > 0. \end{cases} \tag{62}$$

Now, one can prove that as long as $C^*$ is not moving too fast, $C$ converges to $C^*$ in mean square.

**Lemma B.5.** *Let the dynamics of $C^*$ be a drifted Brownian motion: $dC^* = \mu dt + s dW$, where $W$ is a Brownian motion with variance $s^2$. If there exists $c_0 > 0$ such that $\frac{\text{Tr}[\Sigma(\theta^*)A_+^2]}{(\theta^*)^\top A_+^2 \theta^*} \geq c_0$ and $\frac{\text{Tr}[\Sigma(\theta^*)A_-^2]}{(\theta^*)^\top A_-^2 \theta^*} > c_0$,*

$$\mathbb{E}(C - C^*)^2 \leq \frac{2\mu^2 + s^2}{2\sigma^4 c_0^2} = O(1). \tag{63}$$

*Proof.* By assumption,

$$\frac{\dot{C}}{C - C^*} \leq -\sigma^2 c_0. \tag{64}$$

Let us first focus on the case when $C - C^* > 0$. By the definition of $C^*$ and Ito's lemma,

$$d(C - C^*) \leq -\sigma^2 c_0 (C - C^*)dt - \mu dt - s dW. \tag{65}$$

Let $Z = e^{\sigma^2 c_0 t}(C - C^*)$, we obtain that

$$dZ = e^{\sigma^2 c_0 t} d(C - C^*) + \sigma^2 c_0 e^{\sigma^2 c_0 t}(C - C^*)dt \tag{66}$$

$$\leq -\mu e^{\sigma^2 c_0 t} dt - s e^{\sigma^2 c_0 t} dW. \tag{67}$$

Its solution is given by

$$Z \leq -\frac{\mu e^{\sigma^2 c_0 t}}{\sigma^2 c_0} - s \int e^{\sigma^2 c_0 t} dW. \tag{68}$$

Alternatively, if $C - C^* < 0$, we let $Z = e^{\sigma^2 c_0 t}(C^* - C)$, and obtain

$$Z \leq \frac{\mu e^{\sigma^2 c_0 t}}{\sigma^2 c_0} + s \int e^{\sigma^2 c_0 t} dW. \tag{69}$$

Thus,

$$\mathbb{E}[Z^2] \leq \frac{\mu^2 e^{2\sigma^2 c_0 t}}{\sigma^4 c_0^2} + s^2 \int e^{2\sigma^2 c_0 t} dt \tag{70}$$

$$= \frac{\mu^2 e^{2\sigma^2 c_0 t}}{\sigma^4 c_0^2} + \frac{s^2 e^{2\sigma^2 c_0 t}}{2\sigma^2 c_0}. \tag{71}$$

where we have used Ito's isometry in the first line. By construction,

$$\mathbb{E}[(C - C^*)^2] \leq \frac{2\mu^2 + s^2}{2\sigma^4 c_0^2}. \tag{72}$$

The proof is complete. $\square$

Let $\to_p$ denote convergence in probability. One can now prove the following theorem, the convergence of the relative distance to zero in probability.

**Theorem B.6.** *Let the assumptions be the same as Lemma.* (B.5). *Then, if $s = \mu = 0$, $\mathbb{E}[(C - C^*)^2] \to 0$. Otherwise,*

$$\frac{(C - C^*)^2}{(C^*)^2} \to_p 0. \tag{73}$$

*Proof.* By Lemma B.5 and Markov's inequality:

$$\Pr(|C(t) - C^*(t)| > t^{1/4}) \to 0. \tag{74}$$

Now, consider the distribution of $(C^*)^2$. Because $C^*$ is a Gaussian variable with mean $\mu t$ and variance $s^2 t$, we have that

$$\Pr(|C^*| > \sqrt{t}) \to 1. \tag{75}$$

Now,

$$\Pr(|C(t) - C^*(t)|/|C^*| > t^{-1/4}) \geq \Pr(|C(t) - C^*(t)| > t^{1/4} \& |C^*| < \sqrt{t}) \tag{76}$$

$$\geq \max\left(0, \Pr(|C(t) - C^*(t)| > t^{1/4}) + \Pr(|C^*| < \sqrt{t}) - 1\right) \tag{77}$$

$$\to 0, \tag{78}$$

where we have used the Frechet inequality in the second line. This finishes the proof. $\square$

### B.4 Proof of Proposition 5.1

*Proof.* Recall that $U = (\tilde{u}_1, \cdots, \tilde{u}_{d_2})^\top \in \mathbb{R}^{d_2 \times d}$, $W = (\tilde{w}_1, \cdots, \tilde{w}_{d_0}) \in \mathbb{R}^{d \times d_0}$, where $\tilde{u}_i, \tilde{w}_i \in \mathbb{R}^d$. $\theta = \text{vec}(U, W) = (\tilde{u}_1^\top, \cdots, \tilde{u}_{d_2}^\top, \tilde{w}_1^\top, \cdots, \tilde{w}_{d_0}^\top)^\top \in \mathbb{R}^{(d_2 + d_0)d}$.

$$\dot{C} = \eta \text{Tr}(\Sigma(\theta) \nabla_{\theta\theta}^2 C),$$

For $\nabla_{\theta\theta}^2 C$, it holds that

$$\nabla_{\tilde{u}_i, \tilde{u}_j}^2 C = \begin{cases} B, & i = j; \\ 0, & \text{otherwise.} \end{cases}, \quad \nabla_{\tilde{w}_i, \tilde{w}_j}^2 C = \begin{cases} -B, & i = j; \\ 0, & \text{otherwise.} \end{cases} \quad \nabla_{\tilde{u}_i, \tilde{w}_j}^2 C = 0.$$

Therefore,

$$\text{Tr}[\Sigma(\theta)\nabla_{\theta\theta}^2 C] = \sum_{i=1}^n \text{Tr}\left[\nabla_\theta \ell_i \nabla_\theta \ell_i^\top \nabla_{\theta\theta}^2 C\right] = \sum_{i=1}^n \left(\sum_{k=1}^{d_2} \text{Tr}\left[\nabla_{\tilde{u}_k} \ell_i \nabla_{\tilde{u}_k} \ell_i^\top B\right] - \sum_{l=1}^{d_0} \text{Tr}\left[\nabla_{\tilde{w}_l} \ell_i \nabla_{\tilde{w}_l} \ell_i^\top B\right]\right)$$

$$= \sum_{k=1}^{d_2} \text{Tr}[\Sigma(\tilde{u}_k) B] - \sum_{l=1}^{d_0} \text{Tr}[\Sigma(\tilde{w}_l) B].$$

The proof is complete. $\square$

### B.5 Proofs of Proposition 5.3

*Proof.* The loss function is

$$\ell = \|UWx - y\|^2 + \gamma(\|U\|_F^2 + \|W\|_F^2).$$

Let us adopt the following notation: $U = (\tilde{u}_1, \cdots, \tilde{u}_{d_y})^\top \in \mathbb{R}^{d_y \times d}$, $W = (\tilde{w}_1, \cdots, \tilde{w}_{d_x}) \in \mathbb{R}^{d \times d_x}$, where $\tilde{u}_i, \tilde{w}_i \in \mathbb{R}^d$. $\theta = \text{vec}(U, W) = (\tilde{u}_1^\top, \cdots, \tilde{u}_{d_y}^\top, \tilde{w}_1^\top, \cdots, \tilde{w}_{d_x}^\top)^\top \in \mathbb{R}^{(d_x + d_y)d}$.

Due to

$$\nabla_{\tilde{u}_i} \ell = Wx(\tilde{u}_i^\top Wx - y_i) + 2\gamma \tilde{u}_i, \quad \forall i \in [d_y];$$

$$\nabla_{\tilde{w}_j} \ell = \sum_{i=1}^{d_y} \tilde{u}_i x_j \left(\tilde{u}_i^\top Wx - y_i\right) + 2\gamma \tilde{w}_j, \quad \forall j \in [d_x];$$

the diagonal blocks of the Hessian $\nabla_{\theta\theta}^2 \ell$ have the following form:

$$\nabla_{\tilde{u}_i, \tilde{u}_i}^2 \ell = Wxx^\top W^\top + 2\gamma I, \quad \forall i \in [d_y];$$

$$\nabla_{\tilde{w}_j, \tilde{w}_j}^2 \ell = x_j^2 \sum_{i=1}^{d_y} \tilde{u}_i \tilde{u}_i^\top + 2\gamma I, \quad \forall j \in [d_x].$$

The trace of the Hessian is a good metric of the local stability of the GD and SGD algorithm because the trace upper bounds the largest Hessian eigenvalue. For this loss function, the trace of the Hessian of the empirical risk is

$$S(U, W) := \text{Tr}[\nabla^2_{\theta\theta}\ell - 2\gamma I]$$

$$= \sum_{i=1}^{d_y} \text{Tr}[Wxx^\top W^\top] + \sum_{j=1}^{d_x} \text{Tr}[x_j^2 \sum_{i=1}^{d_y} \tilde{u}_i \tilde{u}_i^\top]$$

$$= d_y \text{Tr}[W\Sigma_x W^\top] + \|U\|_F^2 \text{Tr}[\Sigma_x] = d_y \|W\Sigma_x^{1/2}\|_F^2 + \|U\|_F^2 \text{Tr}[\Sigma_x],$$

where $\Sigma_x = xx^\top$. $\qquad\square$

## C  Proof of Theorem 5.2

*Proof.* First, we split $U$ and $W$ like $U = (u_1, \cdots, u_d) \in \mathbb{R}^{d_y \times d}$ and $W = (w_1^\top, \cdots, w_d^\top)^\top \in \mathbb{R}^{d \times d_x}$. The quantity under consideration is $C_B = \text{Tr}[UBU^\top] - \text{Tr}[W^\top BW]$ for an arbitrary symmetric matrix $B$. What will be relevant to us is the type of $B$ that is indexed by two indices $k$ and $l$ such that

$$\begin{cases} B_{ij}^{(k,l)} = B_{ji}^{(k,l)} = 1 & \text{if } i = k \text{ and } j = l \text{ or } i = l \text{ and } j = k; \\ B_{ij}^{(k,l)} = 0 & \text{otherwise.} \end{cases} \tag{79}$$

Specifically, for $k, l \in [d]$, we select $B_{i,j}^{(k,l)} = \delta_{i,k}\delta_{j,l} + \delta_{i,l}\delta_{j,k}$ in $C_B$. With this choice, for an arbitrary pair of $k$ and $l$,

$$C_{B^{(k,l)}} = u_k^\top u_l - w_k^\top w_l.$$

and

$$W^\top B^{(k,l)} W = w_k w_l^\top + w_l w_k^\top, \tag{80}$$

$$UB^{(k,l)}U^\top = u_k u_l^\top + u_l u_k^\top. \tag{81}$$

Therefore,

$$\mathbb{E} \sum_{i=1}^{d_y} \text{Tr}\left[\Sigma(\tilde{u}_i)B^{(k)}\right] = \mathbb{E} \sum_{i=1}^{d_y} (\tilde{u}_i^\top Wx - y_i)^2 \text{Tr}\left[Wxx^\top W^\top B^{(k)}\right] \tag{82}$$

$$= \mathbb{E}\left[\|r\|^2 \text{Tr}[Wxx^\top W^\top B^{(k,l)}]\right] \tag{83}$$

$$= \text{Tr}\left[\mathbb{E}[\|r\|^2 xx^\top]W^\top B^{(k)}W\right] \tag{84}$$

$$= \text{Tr}[\Sigma'_W(w_k w_l^\top + w_l w_k^\top)] \tag{85}$$

$$= 2w_k^\top \Sigma'_w w_l \tag{86}$$

where we have defined $r_i = \tilde{u}_i^\top Wx - y_i$ and $\Sigma'_W = \mathbb{E}[\|r\|^2 xx^\top]$.

Likewise, we have that

$$\mathbb{E} \sum_{j=1}^{d_x} \text{Tr}\left[\Sigma(\tilde{w}_j)B^{(k)}\right] = \mathbb{E} \sum_{j=1}^{d_x} x_j^2 \text{Tr}\left[\left(\sum_{i=1}^{d_y} \tilde{u}_i(\tilde{u}_i^\top Wx - y_i)\right)\left(\sum_{i=1}^{d_y}(\tilde{u}_i^\top Wx - y_i)\tilde{u}_i^\top\right)B^{(k)}\right]$$

$$= \text{Tr}\left[\mathbb{E}[\|x\|^2 U^\top rr^\top UB^{(k,l)}]\right]$$

$$= \text{Tr}[\Sigma'_U UB^{(k,l)}U^\top]$$

$$= 2u_k^\top \Sigma'_u u_l.$$

where we have defined $\Sigma'_u = \mathbb{E}[\|x\|^2 rr^\top]$. Therefore, we have found that for arbitrary pair of $k$ and $l$

$$\dot{C}_{B^{(k,l)}} = -2\gamma(u_k^\top u_l - w_k^\top w_l) + 2\eta(w_k^\top \Sigma'_w w_l - u_k^\top \Sigma'_u u_l). \tag{87}$$

The fixed point of this dynamics is:

$$w_k^\top \Sigma_w w_l = u_k^\top \Sigma_u u_l. \tag{88}$$

where $\Sigma_w = \eta\Sigma'_w + \gamma I$ and $\Sigma_U = \eta\Sigma'_u + \gamma I$. Because this holds for arbitrary $k$ and $l$, the equation can be written in a matrix form:

$$W\Sigma_w W^\top = U^\top \Sigma_u U. \tag{89}$$

Let $V = UW$. To show that a solution exists for an arbitrary $V$. Let $W' = W\sqrt{\Sigma_w}$ and $U' = \sqrt{\Sigma_u}U$, which implies that

$$U'W' = \sqrt{\Sigma_u}V\sqrt{\Sigma_w} := V', \tag{90}$$

and

$$W'(W')^\top = (U')^\top U'. \tag{91}$$

Namely, $U'$ and $(W')^\top$ must have the same right singular vectors and singular values. This gives us the following solution. Let $V' = LSR$ be the singular value decomposition of $V'$, where $L$ and $R$ are orthogonal matrices an $S$ is a positive diagonal matrix. Then, for an arbitrary orthogonal matrix $F$, the following choice of $U'$ and $W'$ satisfies the two conditions:

$$\begin{cases} U' = L\sqrt{S}F; \\ W' = F^\top\sqrt{S}R. \end{cases} \tag{92}$$

This finishes the proof. $\qquad\square$

## D   Discrete-Time GD and SGD

In fact, our results hold in a similar form for discrete-time GD *and* SGD. Let us focus on the exponential symmetries with the symmetric matrix $A$.

The following equation holds with probability 1:

$$0 = \nabla_\theta \ell(\theta, z) \cdot A\theta. \tag{93}$$

For discrete-time SGD, it is notationally simpler and without loss of generality to regard $\ell(\theta)$ as the minibatch-averaged loss, which is the notation we adopt here. This is because if a symmetry holds for every per-sample loss, then it must also hold for every empirical average of these per-sample losses.

The dynamics of SGD gives

$$\Delta\theta_t = -\eta\nabla_\theta\ell(\theta_t, z). \tag{94}$$

This means that

$$\Delta\theta_t \cdot J(\theta) = 0. \tag{95}$$

Therefore, we have that

$$\Delta C_t = 2\Delta\theta_t^\top A\theta_t + \Delta\theta_t^\top A\Delta\theta_t \tag{96}$$

$$= \Delta\theta_t^\top A\Delta\theta_t. \tag{97}$$

Therefore,

$$\Delta C_t = \eta^2 \mathrm{Tr}[\tilde{\Sigma}_d(\theta)A], \tag{98}$$

where $\tilde{\Sigma}_d(\theta) = \nabla_\theta\ell(\theta)\nabla_\theta^\top\ell(\theta)$ is by definition PSD. Already, note the similarity between Eq. (98) and its continuous-time version. The qualitative discussions carry over: if $A$ is PSD, $C_t$ increases monotonically.

Now, while the first-order terms in $\eta$ also vanish in the r.h.s, the problem is that the r.h.s. becomes stochastic because $\Sigma_d(\theta_t)$ is different for every time step. However, one can still analyze the expected flow and show that the expected flow (over the sampling of minibatches) is zero at a unique point in a way similar to the continuous-time limit of the problem. Therefore, we define

$$G_d(\theta_t) = \mathbb{E}_z[\Delta C_t], \tag{99}$$

$$\Sigma_d(\theta_t) = \mathbb{E}_z[\tilde{\Sigma}_d]. \tag{100}$$

We can now prove the following theorem. Note that this theorem applies for any batch size, and so it applies to both SGD and GD.

**Theorem D.1.** *(Discrete-time fixed point theorem of SGD.) Let the per-sample loss satisfy the $A$ exponential symmetry and $\theta_\lambda := \exp[\lambda A]\theta$. Then, for any $\theta$ and any $\gamma \in \mathbb{R}$,*

*(1) $G_d(\theta_\lambda)$ is a monotonically decreasing function of $\lambda$;*

*(2) there exists a $\lambda^* \in \mathbb{R} \cup \{\pm\infty\}$ such that $G_d(\theta_\lambda) = 0$;*

*(3) in addition, if $\mathrm{Tr}[\Sigma_d(\theta)A] \neq 0$ or $\mathrm{Tr}[\theta\theta^\top A] \neq 0$, $\lambda^*$ is unique and $G_d(\theta_\lambda)$ is strictly monotonic;*

*(4) in addition to (3), if $\Sigma_d(\theta)$ is differentiable, $\lambda^*(\theta)$ is a differentiable function of $\theta$.*

*Proof.* Similarly, let us establish the relationship between $\nabla\ell(\theta)$ and $\nabla\ell(\exp(\lambda A))$. By the definition of the exponential symmetry, we have that for an arbitrary $\lambda$,

$$\ell(\theta) = \ell(e^{\lambda A}\theta). \tag{101}$$

Taking the derivative of both sides, we obtain that

$$\nabla_\theta \ell(\theta) = e^{\lambda A}\nabla_{\theta_\lambda}\ell(\theta_\lambda), \tag{102}$$

The standard result of Lie groups shows that $e^{\lambda A}$ is full-rank and symmetric, and its inverse is $e^{-\lambda A}$. Therefore, we have

$$e^{-\lambda A}\nabla_\theta\ell(\theta) = \nabla_{\theta_\lambda}\ell(\theta_\lambda). \tag{103}$$

Now, we apply this relation to the trace of interest. By definition,

$$\Sigma_d(\theta_\lambda) = \mathbb{E}[\nabla_{\theta_\lambda}\ell(\theta_\lambda)\nabla_{\theta_\lambda}^\top\ell(\theta_\lambda)] \tag{104}$$

$$= e^{-\lambda A}\Sigma_d(\theta)e^{-\lambda A}. \tag{105}$$

Because $e^{\lambda A}$ is a function of $A$, it commutes with $A$. Therefore,

$$\mathrm{Tr}[\Sigma_d(\theta_\lambda)A] = \mathrm{Tr}[e^{-\lambda A}\Sigma_d(\theta)e^{-\lambda A}A] \tag{106}$$

$$= \mathrm{Tr}[e^{-2\lambda A}\Sigma_d(\theta)A]. \tag{107}$$

Similarly, the regularization term is

$$\gamma\theta_\lambda^\top A\theta_\lambda = \gamma\mathrm{Tr}[\theta\theta^\top A e^{2\lambda A}] \tag{108}$$

Now, if $\mathrm{Tr}[\Sigma_d(\theta)A] = \theta^\top A\theta = 0$, we have already proved item (2) of the theorem. Therefore, let us consider the case when either (or both) $\mathrm{Tr}[\Sigma_d(\theta)A] \neq 0$ or $\theta^\top A\theta \neq 0$

Without loss of generality, we assume $\gamma \geq 0$, and the case of $\gamma < 0$ follows an analogous proof. In such a case, we can write the trace in terms of the eigenvectors $n_i$ of $A$:

$$-\gamma\theta_\lambda^\top A\theta_\lambda + \eta\mathrm{Tr}[\Sigma_d(\theta_\lambda)A] = \underbrace{\eta\sum_{\mu_i>0}e^{-2\lambda|\mu_i|}|\mu_i|\sigma_i^2 + \gamma\sum_{\mu_i<0}e^{-2\lambda|\mu_i|}|\mu_i|\tilde{\theta}_i^2}_{I_1(\lambda)} - \underbrace{\left(\eta\sum_{\mu_i<0}e^{2\lambda|\mu_i|}|\mu_i|\sigma_i^2 + \gamma\sum_{\mu_i>0}e^{2\lambda|\mu_i|}|\mu_i|\tilde{\theta}_i^2\right)}_{I_2(\lambda)}$$

$$=: I(\lambda),$$

where $\mu_i$ is the $i$-th eigenvalue of $A$, $\tilde{\theta}_i = (n_i^\top\theta_i)^2$, $\sigma_i^2 = n_i^\top\Sigma_d(\theta)n_i \geq 0$ is the norm of the projection of $\Sigma_d$ in this direction.

By definition, $I_1$ is either a zero function or strictly monotonically increasing function with $I_1(-\infty) = +\infty, I_1(+\infty) = 0$ Likewise, $I_2$ is either a zero function or a strictly monotonically increasing function with $I_2(-\infty) = 0, I_2(+\infty) = +\infty$. By the assumption $\mathrm{Tr}(\Sigma_d(\theta)A) \neq 0$ or $\mathrm{Tr}(\theta\theta^\top A) \neq 0$, we have that at least one of $I_1$ and $I_2$ must be a strictly monotonic function.

- If $I_1$ or $I_2$ is zero, we can take $\lambda$ to be either $+\infty$ or $-\infty$ to satisfy the condition.

- If both $I_1$ and $I_2$ are nonzero, then $I = I_1 - I_2$ is a strictly monotonically decreasing function with $I(-\infty) = +\infty$ and $I(+\infty) = -\infty$. Therefore, there must exist only a unique $\lambda^* \in \mathbb{R}$ such that $I(\lambda^*) = 0$.

The proof of (4) follows from the Implicit Function Theorem, as in the continuous-time case. $\square$

The final question is this: what does it mean for $\theta$ to reach a point where $G_d(\theta) = 0$? An educated guess can be made: the fluctuation in $C$ does not vanish, but the flow takes $C$ towards this vanishing flow point – something like a Brownian motion trapped in a local potential well [37]. However, it is difficult to say more without specific knowledge of the systems.

# E  Proof of Theorem 5.4

We first prove the following theorem, which applies to an arbitrary parameter that are not necessarily local minima of of the loss.

**Theorem E.1.** *Let $r = W_D \cdots W_1 x - y$, $\xi_{i+1} := W_D \cdots W_{i+2}$ and $h_i := W_{i-1} \cdots W_1$. For all layer $i$, the equilibrium is achieved at*

$$W_{i+1}^\top \xi_{i+1}^\top C_0^i \xi_{i+1} W_{i+1} = W_i h_i C_1^i h_i^\top W_i^\top, \tag{109}$$

*where $C_0^i = \mathbb{E}[\|h_i x\|^2 r r^\top], C_1^i := \mathbb{E}[\|\xi_{i+1}^\top r\|^2 x x^\top]$. Or equivalently,*

$$\xi_i^\top C_0^i \xi_i = h_{i+1} C_1^i h_{i+1}^\top. \tag{110}$$

*Proof.* By Proposition 5.1,

$$\frac{d}{dt} C_B^i = \sigma^2 \left( \mathbb{E}\mathrm{Tr}\left[ \frac{\partial \ell}{\partial W_{i+1}} B \left( \frac{\partial \ell}{\partial W_{i+1}} \right)^\top \right] - \mathbb{E}\mathrm{Tr}\left[ \left( \frac{\partial \ell}{\partial W_i} \right)^\top B \frac{\partial \ell}{\partial W_i} \right] \right). \tag{111}$$

The derivatives are

$$\frac{\partial \ell}{\partial W_{i+1}} = \xi_{i+1}^\top r (W_i h_i x)^\top, \tag{112}$$

$$\frac{\partial \ell}{\partial W_i} = \xi_{i+1}^\top W_{i+1}^\top r (h_i x)^\top. \tag{113}$$

Therefore, the two terms on R.H.S of Eq. (111) are given by

$$\mathbb{E}\mathrm{Tr}\left[ \frac{\partial \ell}{\partial W_{i+1}} B \left( \frac{\partial \ell}{\partial W_{i+1}} \right)^\top \right] = \mathbb{E}\mathrm{Tr}[\xi_{i+1}^\top r (W_i h_i x)^\top B (W_i h_i x) r^\top \xi_{i+1}],$$

$$= \mathbb{E}\|\xi_{i+1}^\top r\|^2 \mathrm{Tr}[h_i x x^\top h_i^\top W_i^\top B W_i] \tag{114}$$

$$\mathbb{E}\mathrm{Tr}\left[ \left( \frac{\partial \ell}{\partial W_i} \right)^\top B \frac{\partial \ell}{\partial W_i} \right] = \mathrm{Tr}[W_{i+1}^\top \xi_{i+1}^\top r (h_i x)^\top B (h_i x) r^\top \xi_{i+1} W_{i+1}]$$

$$= \mathbb{E}[\|h_i x\|^2 \mathrm{Tr}[W_{i+1}^\top \xi_{i+1}^\top r r^\top \xi_{i+1} W_{i+1} B]]. \tag{115}$$

Because the matrix $B$ is arbitrary, we can let $B_{i,j} = \delta_{i,k}\delta_{j,l} + \delta_{i,l}\delta_{j,k}$. Then, the two terms become

$$\mathbb{E}\mathrm{Tr}\left[ \frac{\partial \ell}{\partial W_{i+1}} B \left( \frac{\partial \ell}{\partial W_{i+1}} \right)^\top \right] = 2\mathbb{E}[\|\xi_{i+1}^\top r\|^2 \tilde{w}_{i,k}^\top h_i x x^\top h_i^\top \tilde{w}_{i,l}], \tag{116}$$

$$\mathbb{E}\mathrm{Tr}\left[ \left( \frac{\partial \ell}{\partial W_i} \right)^\top B \frac{\partial \ell}{\partial W_i} \right] = 2\mathbb{E}[\|h_i x\|^2 \tilde{w}_{i+1,k}^\top \xi_{i+1}^\top r r^\top \xi_{i+1} \tilde{w}_{i+1,l}]. \tag{117}$$

Here, we define the vectors $W_i = (\tilde{w}_{i,1}^\top, \cdots, \tilde{w}_{i,d}^\top)^\top$ and $W_{i+1} = (\tilde{w}_{i+1,1}, \cdots, \tilde{w}_{i+1,d})$. Because Eq. (116) and (117) hold for arbitrary $k, l$, we have

$$\mathbb{E}\mathrm{Tr}\left[ \frac{\partial \ell}{\partial W_{i+1}} B \left( \frac{\partial \ell}{\partial W_{i+1}} \right)^\top \right] = 2W_i h_i \mathbb{E}[\|\xi_{i+1}^\top r\|^2 x x^\top] h_i^\top W_i^\top, \tag{118}$$

$$\mathbb{E}\mathrm{Tr}\left[ \left( \frac{\partial \ell}{\partial W_i} \right)^\top B \frac{\partial \ell}{\partial W_i} \right] = 2W_{i+1}^\top \xi_{i+1}^\top \mathbb{E}[\|h_i x\|^2 r r^\top] \xi_{i+1} W_{i+1}. \tag{119}$$

For Eq. (111) to be 0, we must have

$$W_i h_i \mathbb{E}[\|\xi_{i+1}^\top r\|^2 x x^\top] h_i^\top W_i^\top = W_{i+1}^\top \xi_{i+1}^\top \mathbb{E}[\|h_i x\|^2 r r^\top] \xi_{i+1} W_{i+1}, \tag{120}$$

which is Eq. (109). The proof is complete. $\qquad\square$

We are now ready to prove Theorem 5.4.

*Proof.* It suffices to specialize Theorem E.1 to the global minimum. At the global minimum, we can define

$$r = W_D^* \cdots W_1^* x - y = \epsilon. \tag{121}$$

Then, Eq. (109) can be written as

$$W_{i+1}^\top \frac{W_{i+2}^\top \cdots W_D^\top \Sigma_\epsilon W_D \cdots W_{i+2}}{\mathrm{Tr}[W_{i+2}^\top \cdots W_D^\top \Sigma_\epsilon W_D \cdots W_{i+2}]} W_{i+1} = W_i \frac{W_{i-1} \cdots W_1 \Sigma_x W_1^\top \cdots W_{i-1}^\top}{\mathrm{Tr}[W_{i-1} \cdots W_1 \Sigma_x W_1^\top \cdots W_{i-1}^\top]} W_i^\top. \tag{122}$$

To solve Eq. (122), we substitute $W_D$ and $W_1$ with $W_1' = W_1 \sqrt{\Sigma_x}$ and $W_D' = \sqrt{\Sigma_\epsilon} W_D$, which transform Eq. (122) into

$$W_{i+1}^\top \frac{W_{i+2}^\top \cdots W_D'^\top W_D' \cdots W_{i+2}}{\mathrm{Tr}[W_{i+2}^\top \cdots W_D'^\top W_D' \cdots W_{i+2}]} W_{i+1} = W_i \frac{W_{i-1} \cdots W_1' W_1'^\top \cdots W_{i-1}^\top}{\mathrm{Tr}[W_{i-1} \cdots W_1' W_1'^\top \cdots W_{i-1}^\top]} W_i^\top. \tag{123}$$

The global minimum condition can be written as

$$W_D' W_{D-1} \cdots W_2 W_1' = \sqrt{\Sigma_\epsilon} V \sqrt{\Sigma_x} := V'. \tag{124}$$

Then, we can decompose the matrices $W_1', \cdots, W_D'$ as

$$W_D' = L \Sigma_D U_{D-1}^\top, \ W_i = U_i \Sigma_i U_{i-1}^\top (i \neq 1, D), \ W_1' = U_1 \Sigma_1 R, \tag{125}$$

where $\Sigma_D, \cdots, \Sigma_1 \in \mathbb{R}^{d \times d}$, $L \in \mathbb{R}^{d_y \times d}$, $U_i \in \mathbb{R}^{d_i \times d}$, $R \in \mathbb{R}^{d \times d_x}$ with $d := \mathrm{rank}(V')$ and arbitrary $d_i$. The matrices $U_i$ satisfy $U_i^\top U_i = I_{d \times d}$. By substituting the decomposition into Eq. (123), we have

$$\frac{\Sigma_{i+1} \cdots \Sigma_D \Sigma_D \cdots \Sigma_{i+1}}{\mathrm{Tr}[\Sigma_{i+2} \cdots \Sigma_D \Sigma_D \cdots \Sigma_{i+2}]} = \frac{\Sigma_i \cdots \Sigma_1 \Sigma_1 \cdots \Sigma_i}{\mathrm{Tr}[\Sigma_{i-1} \cdots \Sigma_1 \Sigma_1 \cdots \Sigma_{i-1}]}. \tag{126}$$

Since these diagonal matrices commute with each other, we can see $\Sigma_i = c I_{d \times d}$. Then we move on to fix the parameter $c$. By taking $i = 1$ and $i = D - 1$ in Eq. (126), we obtain

$$\frac{\Sigma_2^2 \cdots \Sigma_D^2}{\mathrm{Tr}[\Sigma_3^2 \ldots \Sigma_D^2]} = c^2 \frac{\Sigma_D^2}{\mathrm{Tr}[\Sigma_D^2]} = \frac{\Sigma_1^2}{d}, \tag{127}$$

$$\frac{\Sigma_D^2}{d} = \frac{\Sigma_1^2 \cdots \Sigma_{D-1}^2}{\mathrm{Tr}[\Sigma_1^2 \ldots \Sigma_{D-1}^2]} = c^2 \frac{\Sigma_1^2}{\mathrm{Tr}[\Sigma_1^2]}, \tag{128}$$

where $d$ represents the dimension of the learning space. By taking trace to both sides of Eqs. (127) and (128), we can see $\mathrm{Tr}[\Sigma_1^2] = \mathrm{Tr}[\Sigma_D^2]$ and hence, $\Sigma_1 = \Sigma_D$. The parameter $c$ is given by

$$c = \sqrt{\frac{\mathrm{Tr}[\Sigma_1^2]}{d}}. \tag{129}$$

With the SVD decomposition $V' = LS'R$, we have

$$\Sigma_1^2 c^{D-2} = S'. \tag{130}$$

Therefore, the solutions for $c$ and $\Sigma_1$ are

$$c = \left(\frac{\mathrm{Tr}S'}{d}\right)^{1/D}, \ \Sigma_1 = \frac{\sqrt{S'}}{c^{(D-2)/2}} = \left(\frac{d}{\mathrm{Tr}S'}\right)^{(D-2)/2D} \sqrt{S'}. \tag{131}$$

The scaling of the diagonal matrices are shown as

$$\mathrm{Tr}[\Sigma_1^2] = d^{1-2/D} (\mathrm{Tr}S')^{2/D}, \ \mathrm{Tr}[\Sigma_i^2] = (\mathrm{Tr}S')^{2/D} d^{1-2/D} = \mathrm{Tr}[\Sigma_1^2]. \tag{132}$$

The proof is complete. $\qquad \square$

