# OpenReview forum: "Parameter Symmetry and Noise Equilibrium of Stochastic Gradient Descent"
_NeurIPS.cc/2024/Conference — NeurIPS 2024 poster_

### Official Review · Reviewer_QY9h · 2024-06-20

**Soundness:** 2
**Presentation:** 3
**Contribution:** 4
**Rating:** 7
**Confidence:** 4

**Summary:**

The authors study the effect of exponential symmetries on the learning dynamics of SGD. They establish the theorem that every exponential symmetry implies the existence of a unique and attractive fixed point in SGD dynamics, which could explain phenomena in machine learning such as matrix factorization and progressive sharpening or flattening.

**Strengths:**

The paper studies the dynamics of SGD. The authors prove a novel theorem that relates continuous symmetry to the attractive fixed point in SGD. The theorem is quite general and can have significant potential impacts on the field. The authors apply their theorem to matrix factorization, offering strong analytical and experimental support. And by my knowledge, this is a novel contribution.

**Weaknesses:**

1.	C is not guaranteed to exist. (Line 70)
2.	Usually, the stability of the learning is determined by the largest eigenvalue of the Hessian instead of the trace as discussed in the paper. The trace may not be an upper bound on the largest eigenvalue as it can contain negative eigenvalues.
3.	I think it’s missing dependence on $\gamma$ in proposition 5.3.
4.	The authors proved that in deep linear networks, the noise will lead to a balanced norm for all intermediate layers. This is an important theorem, but it lacks empirical verification. It is important to verify that the empirics actually match the theorem, which would strengthen the paper by a lot. Such an experiment should not be hard to run.
5.	There is a lack of clarity in how the analyses in Section 5.4 (Equation 24) relate to Figure 4.

**Questions:**

1.	In line 124-128, the authors state that the SGD dynamics can be decomposed into two independent parts. Why does equation (5) have no effect on the loss? I would expect the noise will result in fluctuation of the loss at the end of training, which means that the decomposition should not hold.
2.	For the double rotation symmetry, why it is hard to write the matrix A explicitly? It seems that A can just be any symmetric matrix as discussed in section 5.1. Also, why do you require A to be symmetric when defining the exponential symmetry? I am wondering if the results can be generalized to a general matrix A.
3.	I don’t understand why one needs to assume C* to be a constant in line 163. By definition, C is a deterministic function that only depends on $\theta$.

**Limitations:**

The authors have included the limitations. For additional points, please see my comments above.

---

> ### Author Rebuttal · Authors · 2024-08-04
>
> Thank you very much for the feedback. Please first see our summary rebuttal above.
>
> **Weaknesses:**
>
> **C is not guaranteed to exist. (Line 70)**
>
> Thanks for this comment. This is a well-known fact in the study of symmetries and the Noether theorem. It is required that $J$ needs to be conservative. We will add a note on this. That being said, for the type of symmetries (Def 4.1) we focus on in our work, $C$ always exists.
>
> **Usually, the stability of the learning is determined by the largest eigenvalue of the Hessian... The trace may not be an upper bound on the largest eigenvalue as it can contain negative eigenvalues.**
>
> Thanks for this question. For (deterministic) GD, the training stability is indeed determined by $\lambda_{\max}(Hess)$. However, this does not apply to SGD. In fact, [1][2] demonstrates that the stability of SGD is impacted by $Tr(Hess)$, rather than $\lambda_{\max}(Hess)$. [1][3] suggest that $Tr(Hess)$ significantly determines the properties of SGD’s solutions.
>
>
> [1] Wu and Su. The Implicit Regularization of Dynamical Stability in Stochastic Gradient Descent. (ICML 2023)
>
> [2] Ma and Ying. On the linear stability of SGD and input-smoothness of neural networks. (NeurIPS 2021)
>
> [3] Li et al. What Happens after SGD Reaches Zero Loss? –A Mathematical Framework. (ICLR 2022)
>
>
> **I think it’s missing dependence on γ in proposition 5.3.**
>
> We will fix it in the revision.  In the experiments, we let $\gamma=0$ and so we did not use the weight decay term.
>
>
> **The authors proved that in deep linear networks, the noise will lead to a balanced norm for all intermediate layers. This is an important theorem, but it lacks empirical verification....**
>
> Thanks for this very constructive feedback. We have added an extensive set of experiments for deep linear nets. We show that for MNIST, a deep linear net (with five hidden layers), the intermediate layers converge to the same norm, whereas the input and output layers are special and move towards different norms. This effect is also shown to be robust across different types of initializations. See Figure 8. In addition, see the additional experiments in Figure 7 and Figure 9.
>
>
> **There is a lack of clarity in how the analyses in Section 5.4 (Equation 24) relate to Figure 4.**
>
> Thanks for this criticism. Section 5.4 (and Eq 24) shows that if a model approximately has a symmetry (for example, when it can be approximated by a model with symmetry), then its solution will be biased towards the noise equilibrium of this symmetry model.
>
> Figure 4 shows exactly this. A tanh network can be approximated by a linear network of the same depth and connectivity (because $tanh(wx) \approx wx$ to leading order). For a linear network, it is true that the noise equilibrium will satisfy $W\Sigma_xW^T = U^T \Sigma_\epsilon U$, and so one would expect that under SGD training, a tanh network will also be biased towards satisfying this relation, and this qualitative prediction is confirmed by Figure 4. We will add this explanation to the revision.
>
> **Questions:**
>
> **In line 124-128, the authors state that the SGD dynamics can be decomposed into two independent parts. Why does equation (5) have no effect on the loss?**
>
> This is a good question. This statement is made assuming that the fluctuation is not significant. This is supported by the experiment in Figure 3  and the mid panel of Figure 2, where, at the end of training, the fluctuation in the converged solution is small.
>
> Also, there is a deeper mathematical reason for this statement. The fluctuation can also be decomposed into the parts in the subspace parallel to the gradient and orthogonal to it. The fluctuation in the orthogonal subspace (which is our main focus of study) will not affect the loss.
>
>
> **For the double rotation symmetry, why it is hard to write the matrix A explicitly?... I am wondering if the results can be generalized to a general matrix A**
>
> Thanks for this question. It is just cumbersome and space-taking to write. First of all, to write it in the form 4.1, one needs to view both matrices (U and W) as vectors, and then the corresponding symmetry matrix will be composed by a sequence of block matrices: A=diag(B,...,B,B^{-1},...,B^{-1}), where each block corresponds to a column or row of $U$ or $W$.
>
> Requiring $A$ to be symmetric is actually to ensure that a Noether charge of the form $\theta^T A \theta$. Extending the result to antisymmetric matrices is highly nontrivial, and we know very little about this type of symmetry. On the one hand, noether charge does exist for some antisymmetric matrix. For an example, consider rotation symmetry for (u,w) in 2d. The antisymmetric matrix is ((0,-1),(1,0)). The symmetry implies the following dynamics for GD: $wdu - udw = 0$. Simplifying the relation leads to (ignoring the sign problem for now) d(log(u) - log(w)) = 0. We have thus constructed a Noether charge for an antisymmetric-matrix symmetry: log(u) - log(w); however, it is no longer in a quadratic form.
>
> On the other hand, we do not know a general formula for them or whether they exist for an arbitrary antisymmetric symmetry, which is an important open problem in the field.
>
> **I don’t understand why one needs to assume $C^\*$ to be a constant in line 163...**
>
> This is a good question. This is because $C^*$ depends on the value of $\theta$ in the nondegenerate subspace, and $\theta$ has a non-zero time derivative in general. To see this mathematically, note that by definition $C^*= C(\theta_{\lambda^*})$, but $\lambda^*=\lambda^*(\theta)$ is a function of $\theta$ as well, and so $C^*$ sometimes depends on time through $\theta$'s dependence on time (which is often non-zero due to gradient or random noise).
>
> Therefore, a good mental and intuitive picture of the process is that there is a run-and-catch game between $C^*$ and $C$: $C^*$ moves around randomly (due to the motion of $\theta$), and $C$ chases it systematically due to symmetry.

---

> > ### Comment · Reviewer_QY9h · 2024-08-09
> >
> > I appreciate the author’s response to my review. The author has addressed most of my concerns and the author also added more empirical experiments to verify the theorems claimed in the paper. All these have strengthened the paper and as a result, I am increasing my score.
> >
> > As a side note, I have also reviewed the comments from the other reviewers. While there is some debate regarding the soundness, precision of language, and overall presentation of the paper, I respectfully disagree with reviewer HyRa in rating the contribution of this paper as ‘poor’. Although some of the related works and discussions are indeed missing (as noted by rhMv), the results in this paper are novel to the best of my knowledge and contribute meaningfully to a better understanding of the role of SGD.

---

### Official Review · Reviewer_HyRa · 2024-07-07

**Soundness:** 2
**Presentation:** 2
**Contribution:** 2
**Rating:** 5
**Confidence:** 3

**Summary:**

This paper considers the relationship between the behavior of SGD for solving empirical risk minimization and the exponential symmetry.

**Strengths:**

The strength of this paper is to study the relationship (Theorem 4.3) between the behavior of SGD for solving empirical risk minimization and the exponential symmetry (Definition 4.1). Moreover, it considers a noise equilibrium (Definition 4.5) and provides explicit solutions of the noise equilibriums for some applications (Section 5).  In addition, it provides some numerical results to support the theoretical analyses presented in this paper.

**Weaknesses:**

The weaknesses are the following. Please see Questions for details.
- The mathematical preliminaries are insufficient.
- The assumptions seem to be strong.
- The explanations of contributions are insufficient.

**Questions:**

- SGD for minimizing $L(\theta) = E_z [\ell (\theta,z)]$ is defined by (1): $d \theta_t = - \nabla L(\theta_t) dt + \sqrt{2 \sigma^2 \Sigma (\theta_t)} d W_t$ using $\sigma^2 = \eta/(2S)$, where $\eta > 0$ is a step-size and $S > 0$ is a batch size. My first question is "Which is a problem considered in this paper: the expected risk minimization or the empirical risk minimization?"  $L$ is defined by using the expectation $E$ (Lines 56-57), however, Line 57 says "the empirical risk function."
- Line 19 says that SGD with $\sigma = 0$ implies GD. I do not understand it, since $\sigma^2 = \eta/(2S) = 0$ iff $\eta = 0$, which contradicts $\eta > 0$.
- Line 21: I do not understand what $\sigma^2 \ll 1$ is.
- Line 58: Please define $\Sigma_v (\theta)$ explicitly.
- Line 93: a --> an
- Eq (7): Is $-4 \gamma$ in (7) replaced with $-2 \gamma$?
- Line 142: I do not understand the definition of "fixed point." Please define "fixed point" explicitly. A fixed point $x$ of a mapping $T \colon R^d \to R^d$ is defined by $x = T(x)$. The definition of "fixed point" considered in this paper seems to be different from a fixed point of a mapping (see Theorem 4.3).
- Eq (8): I do not understand the relationship between $\ell (\theta, z)$ and $\ell$ defined by (8). Why do we consider $\ell$ in (8) not $\ell (\theta, z)$? I think that the main objective of this paper is to minimize $L(\theta) = E_z [\ell (\theta, z)]$.
- Eq (9): We cannot define (9) when $\gamma = 0$. Please define $C(\theta^\star)$ explicitly.
- Eq (11): $\theta'$ is not defined.
- Section 5.1: This section considers a matrix factorization: Given $A$, find $U$ and $W$ such that $A = UW$. The assumption of $A$ seems to be strong such that $A$ is close to the identity matrix $I$ (Lines 204 and 205).
- Proposition 5.1: The assumption "the symmetry (11) holds" (based on $A \approx I$) seems to be strong. Please provide practical examples of empirical risk minimization and SGD satisfying (11).
- Figure 2: There are not definitions of the matrix size, a batch size $S$, $\eta$, and $\ell (\theta,x)$.
- Line 222: $\epsilon$ is not defined.
- Theorem 5.2: Is the theorem based on (11)? If so, please provide some examples of $\dot{C}_B = 0$ under (11).
- Figure 3: There are not definitions of $d_x$, $d_y$, $\epsilon$, $S$, and so on. Moreover, I do not understand why a warm-up with $\eta_\min = 0.001$, $\eta_\max = 0.008$, and the switching steps $5000$ is used. Please justify the setting from the theoretical results.
- ...
- Additional comments: I would like to know new insights that come from the theoretical results in this paper. This is because I am not surprised at the results. For example, can we apply the results to training more practical deep neural networks on practical datasets? Then, can we say new insights for training practical deep neural networks? Moreover, can we estimate appropriate step-size $\eta$, batch size $S$, and hyperparameter $\gamma$ using the theoretical results before implementing SGD? Anyway, I would like to know contributions of this paper.

---

> ### Author Rebuttal · Authors · 2024-08-04
>
> Thank you for your criticisms. We believe that your criticisms are due to misunderstanding of mathematics and the related literature and not paying attention to the details given in the immediate context. We will answer concisely due to space constraints. However, please ask us to elaborate on every point that you need more detail.
>
> **Questions:**
>
> **the expected risk minimization or the empirical risk minimization...**
>
> Thanks for this question, and we apologize for the confusion. In our formalism, it is not essential to differentiate between the population risk and the empirical risk. The loss function can be either the population risk (if the training proceeds with online SGD) or the empirical risk (if the training proceeds with a finite-size dataset). This is because $\mathbb{E}$ denotes the expectation over the distribution of the training data, which can be arbitrary and is problem/algorithm-dependent.
>
> **Line 19...**
>
> We are referring to the limit $S\to \infty$. Within the SDE framework, this limit is the GF/GD algorithm.
>
> **Line 21: I do not understand what σ2≪1 is.**
>
> This is a common notation meaning that "$\sigma$ is far smaller than 1," for example, $\sigma = 0.1$, $0.01$, etc.
>
>
> **Line 58: Please define Σv(θ)...**
>
> This is defined and explained in line 58. Colloquially, $\Sigma_v$ is a shorthand for the gradient covariance for a subset of all parameters $v$.
>
> **Eq (7)...**
>
>  Yes. We will fix it.
>
> **Line 142: I do not understand the definition of "fixed point...**
>
> In the study of dynamical systems (DS), the fixed point is defined differently. In DS, a fixed point is defined for differential equations of the form: $\dot{x} = f(x)$ as the set of $x$ such that $f(x) =0$. See standard textbooks of this field (e.g., Introduction to the modern theory of dynamical systems, by Katok et al.). In this terminology, Def 4.5 is an explicit definition of what "fixed point" means in our context. This definition is related to the fixed point of mappings. Here, the underlying mapping is one step of GD, which maps $\theta_t \to \theta_{t+1}$.
>
>
> **Eq (8): relationship between ℓ(θ,z) and ℓ...**
>
> $\ell$ and $\ell(\theta,z)$ are always the same thing. The $z$ argument is always abbreviated for concision. This has been explained in line 61.
>
> **Eq (9): We cannot define (9) when γ=0.**
>
> This is a good point. It is better to write Eq (9) as $2\gamma C =\sigma^2 Tr[\Sigma A]$, which is now well-defined. We will fix this.
>
> **Please define $C(θ^\*)$ explicitly.**
>
> This quantity is defined in the line 183. $C(\theta^*)= (\theta^*)^T A \theta^*$, and $\theta^*$ is whatever quantity that satisfies $G(\theta^*) = 0$ (whose existence is guaranteed by Theorem 4.3).
>
>
> **Eq (11): θ′ is not defined.**
>
> Thanks for pointing this out. $\theta'$ are the other parameters that are irrelevant to the double rotation symmetry. We will include this in the revision.
>
> **Section 5.1: The assumption of A seems to be strong such that A is close to the identity matrix I....**
>
> We are afraid to say that this is a major misunderstanding of the problem setting of Section 5.1. Here, the goal is to find $U$, $W$ such that $UW=C$ for a fixed matrix $C$ (note that this $C$ is not $A$). This problem has a lot of degeneracy in the solution. For any invertible matrix $A$: $UA$, $A^{-1}W$ is also a solution to $UW=C$. Hence, this problem has "symmetries." The matrix "A" in our original text has to do with these arbitrary invertible transformations, not with the target matrix $C$.
>
> There is no assumption at all. To derive proposition 5.1, one focuses on the type of symmetries that are close to identity: $A \approx I$; this is because these matrices are the generator of the Lie group for these symmetries. This is not an assumption because $A$ can be an arbitrary matrix.
>
> **Proposition 5.1: The assumption "the symmetry (11) holds" (based on A≈I) seems to be strong. Please provide practical examples of empirical risk minimization and SGD satisfying (11).**
>
> See the answer above. This is not an assumption. Eq (11) is satisfied by any (deep) matrix factorization problem.
>
> **Figure 2...**
>
> The experimental details are given in appendix A2. The loss function is MSE and given by Eq (13). The matrix size is 30 by 30, the learning rate is 0.1, and the batch size is 1. We did not specify learning and batch size because the result is highly insensitive to them (as any learning rate and batch size would work and give a similar figure). See Figure 9 of the pdf. We will include these details in the revision.
>
> **Line 222**
>
> $\epsilon$ is a noise on the label. We will fix it.
>
> **Theorem 5.2: Is the theorem based on (11)? If so, please provide some examples of C˙B=0 under (11).**
>
> We are not sure if we understand this question. Theorem 5.2 (and Eq. (14)) enumerates all examples where $\dot{C}_B=0$.
>
> **Figure 3**
>
> They are the same as in Figure 2. $d_x=d_y=30$ and $\epsilon \sim \mathcal{N}(0, I_{30})$ is an independent Gaussian noise. We did not specify learning and batch size because the result is highly insensitive to them. We will clarify this in the revision.
>
> **Moreover, I do not understand why a warm-up with $η_{min}=0.001, η_{max}=0.008$, and the switching steps 5000 is used...**
>
> Like the previous experiment, the result is robust to these choices, and the specific values of these quantities are not essential at all. The specific values are just chosen out of convenience. For example, a switching step of $5000$ is arbitrary, and we observe that choosing it to be any number larger than 1000 will give a similar result. $η_{min}=0.001$ is also arbitrary and nonessential. Choosing it to be any value such that the model does not diverge at initialization will also work.
>
> **...Anyway, I would like to know contributions of this paper.**
>
> Thanks for this important question. This is a good question but admits many different answers. Due to space constraints, we will have to answer this in the comment.

---

> > ### Comment · Reviewer_HyRa · 2024-08-09
> > **Reply to the authors' comments**
> >
> > I also thank you for replies on my comments. I misread the manuscript. I am sorry for my first review. I still have some minor concerns for your replies.
> > - I do not understand why $S \to \infty$ is done. Since $S$ is a fixed batch size, we should replace $S$ with $S_t$?
> > - I do not know why $\sigma$ is less than $1$. Do you mean that $\sigma_t = \eta/(2S_t)$ is less than $1$ for a sufficiently large $t$?

---

> ### Author Response · Authors · 2024-08-05
> **Answer to the Contribution Question**
>
> **I would like to know new insights that come from the theoretical results in this paper. This is because I am not surprised at the results. For example, can we apply the results to training more practical deep neural networks on practical datasets? Then, can we say new insights for training practical deep neural networks... Anyway, I would like to know contributions of this paper..**
>
>
> Essentially, the answer will have to depend on the subjective opinion of the reader. A contribution for researchers in one field may not seem like a contribution for researchers in other fields. The subjectiveness here is clearly reflected in the fact that the other reviewers rate our contribution as "excellent" while you rate it as "poor." How does understanding an insect's brain help us understand the training of diffusion models? It probably does not, and the problem is not with the brains of insects but with this question being ill-posed. Also, keep in mind that NeurIPS is not just a conference for practical deep learning but also a conference for neuroscience and many other interdisciplinary fields of science.
>
> With these caveats in mind, we clarify our contribution from different angles below. In short, our theory is interdisciplinary, requiring knowledge and expertise from many different fields to understand, and it could be significant to these related fields. We believe that our research has the potential to inspire new research in pure mathematics, neuroscience, deep learning, physics, and practices in deep learning.
>
> 1. Pure Mathematics: Lie-group symmetries have been leveraged to solve ordinary differential equations (e.g., see the book Elementary Lie group analysis and ordinary differential equations). However,  leveraging the Lie group to solve stochastic differential equations is a novel concept and tool for pure and applied mathematics, and our result advances our understanding of stochastic systems. The tool we developed is sufficiently general that it may even be applied to problems outside the field of deep learning.
>
> 2. Neuroscience: Understanding how latent representations are formed in animal brains is a fundamental scientific problem in neuroscience. Until now, there has been a very limited understanding of this, and our theory suggests one mechanism of how latent representations form and can inspire neuroscientists to understand biological brains. The fact that symmetry in the solutions can be a core mechanism for latent representation is a novel and potentially surprising message. For example, compare the latent representations in our paper with the biological data in Figure 7B in www.cell.com/neuron/fulltext/S0896-6273(18)30581-6
>
> 3. Theory and Mathematics of Deep Learning: The most important contribution of our work is on the mathematical tool side. We established a new mathematical tool (the formalism of the noise-equilibrium fixed points of SGD), and this formalism has the potential to be applied to many problems in deep learning. For example, we showed its relevance to flattening and sharpening effects during training, the warm-up technique, and the solution of deep matrix factorizations. We emphasize that developing new mathematical tools is an important progress per se. Understanding deep learning is a difficult task, and without new mathematical tools, this would be impossible.
>
> 4. Physics: Symmetry is a fundamental concept of physics. The standard understanding was that when a continuous symmetry exists, a conserved quantity exists during the dynamics. This concept has also played an important role in machine learning; for example, see the well-known result in https://arxiv.org/abs/1806.00900, where the symmetry and its related conserved quantities are leveraged to prove the convergence of the matrix factorization problem. Our theory, however, shows that this picture is no longer right when the dynamics are stochastic -- conserved quantities no longer exist in general, and, in place of them, we now have unique fixed points where the gradient noises must balance. This is a novel and potentially surprising new concept.
>
> 5. Practices in Deep Learning: Lastly, our theory also has implications for the practices of deep learning, although we do not claim this to be a main contribution. The first contribution is that our theory helps us understand when SGD prefers flatter solutions. The concept of flatness is of practical importance (for example, practitioners leverage it to design algorithms: https://arxiv.org/abs/2010.01412). Another practical insight our theory directly offers is the importance of data preprocessing because our theory shows that the data distribution has a major impact on the learning trajectory and the final solution it reaches. For example, see the newly added experiments in Figure 7, where we directly show that the orientation of the data distribution has a direct effect on the balancedness and sharpness of the learned solution.

---

> > ### Comment · Reviewer_HyRa · 2024-08-09
> > **Reply to the authors' comments**
> >
> > I would like to thank the authors for detailed comments. I have read all of them. As a result, I would like to raise my score. However, I still have some concerns. I will write the above form.

---

> ### Author Response · Authors · 2024-08-09
> **reply**
>
> Thanks for the update and additional question.
>
> **Since $S$ is a fixed batch size, we should replace $S$ with $S_t$?**
>
> $S$ is fixed and does not change with time, and so one should not write it as $S_t$. Here, this limit is a mathematical limit of differential equations. One mathematically rigorous way to construct this limit is to define it as a limit of a sequence of differential equations. Operationally, this (roughly) corresponds to running the experiment first with $S=1$. Then, run an independent experiment with $S=2$. Then, with $S=3$, and ad infinitum. During each of these running, $S$ is held fixed, and so does not change with $t$.
>
> **I do not know why $\sigma$ is less than 1.**
>
> This factor is just a hyperparameter determined by the user of SGD. If the user uses a learning rate of $0.1$ and a batch size of $100$, one naturally has $\sigma=0.1/(2\times 100) \ll 1$.

---

### Official Review · Reviewer_rhMv · 2024-07-12

**Soundness:** 3
**Presentation:** 3
**Contribution:** 4
**Rating:** 7
**Confidence:** 4

**Summary:**

The paper takes important steps toward showing that the noise in SGD pushed the dynamics along symmetry directions toward a unique fixed point. This differs in an important way from GD and gradient flow (GF) where one can show that in the presence of continuous symmetries (called exponential symmetries in the paper) there will be conserved quantities $C$, such as layer imbalance. These conserved quantities ensure that runs starting with different values of $C$ end up on different points along a minimum manifold (with some caveats).
This paper shows that SGD noise causes drift toward a fixed point on a minimum manifold.

**Strengths:**

This is potentially a very significant work. This is one of the few works I know which focuses on the effect of SGD noise on the symmetry directions and derives a concrete dynamics for this. I think the approach is at least partly original. Although there are many related works (see weaknesses), this paper is the only one which derives concrete solutions for the noisy dynamics along the symmetry directions.
There are some corners that need to be ironed out. But if the results survive, this work could answer many questions about the implicit bias and generalization in SGD.

**Weaknesses:**

There exist multiple other works addressing the same questions. Two particular ones with very similar claims that the authors may have missed are:
[1] Chen, Feng, Daniel Kunin, Atsushi Yamamura, and Surya Ganguli. "Stochastic collapse: How gradient noise attracts sgd dynamics towards simpler subnetworks." Advances in Neural Information Processing Systems 37 (2023).
[2] Yang, Ning, Chao Tang, and Yuhai Tu. "Stochastic gradient descent introduces an effective landscape-dependent regularization favoring flat solutions." Physical Review Letters 130, no. 23 (2023): 237101.

I ask the authors to clarify the distinction between their work and the above. [1] is also about the effect of SGD noise on the symmetry directions, afaik.
[2] also derives the effective dynamics of SGD as a Langevin equation and shows a result similar to yours, namely that along symmetric minima, the noise drives the system toward the least sharp minimum.

**Questions:**

There are many details int he computations and proofs which I think need to be clarified. One aspect which I think is an error is related to the Lie algebra of the symmetries discussed in the paper. Most derivations in the paper hold for symmetric Lie algebra elements (i.e. generators of scaling and hyperbolic directions in the group) but fail for rotations. A relevant work which studies the parameter space more thoroughly is:
[3] Zhao, Bo, Iordan Ganev, Robin Walters, Rose Yu, and Nima Dehmamy. "Symmetries, flat minima, and the conserved quantities of gradient flow." ICLR (2023).

My questions are as follows:



line 49: ref [40] may be wrong (it doesn't discuss conservation laws) maybe another paper by Zhao is meant: Zhao et al ICLR 2023 "Symmetries, flat minima, and the conserved quantities of gradient flow".
sec 4: important caveat about conserved quantities is that the antiderivative C may not always exist. An important case is when the symmetry is a rotation. (see Zhao et al "symmetries, flat minima ..." ICLR 2023) In this case, the Noether charge is zero.

1. line 78: eq 5: to clarify, $\nabla^T C \nabla \ell(\theta, z) = 0$ is always zero due to eq 2 and 3, right? Is that why both terms in eq 30 (App B) vanish?
However, there are some statements in Appendix B which seem inaccurate to me:
  a. when $\Sigma  = GG^T$, the matrix $G$ need not be square and $G = \sqrt{\Sigma}$ is just one of the possible $G$. What is your argument for this $G$ being the correct $G$?
  b. I don't agree with your statement that "Because $\Sigma$ and $\sqrt{\Sigma}$ share eigenvector..." eq 31 would vanish. Trivial example: the 2x2 matrices $A=diag(1,-1)$ and the identity matrix share eigenvectors, but the vector $v=(1,1)$, $Av = 0$, but $Iv \ne 0$. I think the real reason is that $\Sigma = GG^T$. Doing SVD, $G = USV^T$ and $\Sigma = US^2U^T$. Then eq 30 yields $\nabla^T C U S= 0$. Assuming the covariance is nondegenerate (i.e. $S_{ii}\ne 0$ for all $i$), we must have  $\nabla^T C U = 0$ which also implies $\nabla^T C G = 0$.

2. line 83: at a minimum, some symmetry orbits (i.e. invariant manifold) may collapse to a point, e.g. rotation symmetry in a quadratic loss as in linear regression $\|\theta x -y \|^2$.

3. line 104: Instead of "double rotation" this should be called double linear transformation, becuase the group is not just $O(h)$, but rather $GL(h)$

4. eq 6: As noted above, this Noether charge vanishes for rotations $SO(h)$ because $A$ is anti symmetric and hence $\theta^T A \theta=0$ (see Zhao ICLR 2023).

5. line 114: why? Why the explicit weight decay? Why in defining the symmetry do you ignore $\|\theta\|^2$ which doesn't have rescaling or $GL(n)$ symmetry, only rotation symmetry $O(n)$?

6. Eq 7: what happens to the cross terms of $\theta^t \nabla \ell$ in $\Sigma_\gamma$?

7. line 118: eigenvalues and vectors of A can be complex (e.g. for Lie algebra of rotation), so $n^T$ needs to be $n^\dagger$.

8. line 214-215: I don't understand how you arrive at $\dot{C} = \mathrm{Var}[r]C$. I think you mean when the hidden dimension is 1, not the output, right? I hidden is $d>1$, let $\sigma=\Sigma_{u_i}$, then for $B= E_{kl}+E_{lk}$ we have  $\mathrm{Tr}[\Sigma_{u_i}B] = \sigma_{lk}+\sigma_{kl}$, which are off-diagonal terms in the covariance matrix, unless $k=l$.

9. eq 13: When $\gamma\ne 0$, eq 13 does not have the symmetry (11) for symmetric Lie algebra element $B$. The reason is $\|W\|_F$ is invariant under rotations where $A^{-1} = A^T$. The Lie algebra of rotations (infinitesimal $A = I+\rho B$) has anti-symmetric $B$.

10. line 235: under what condition will the covariances $\Sigma_x$ and $\Sigma_\epsilon$ be isotropic? I don't think this happens often in SGD. I think you need adaptive gradient optimizers or Newton's method to have isotropic $\Sigma_\epsilon$.

11. eq 39,40: $\nabla^T_{\theta_\lambda} \ell = [\nabla_{\theta_\lambda} \ell]^T e^{-\lambda A^T}$, so we don't have (40)  and in (41) there should $e^{-\lambda A^T}A e^{-\lambda A}$.

12. line 531: $Z_++Z_-$ will have complex eigenvalues when $A$ is not symmetric. Diagonalizing $A$, your arguments about $Z_\pm$ only hold for symmetric $A$.

**Limitations:**

There are some gaps in the derivations (see questions) which I hope the authors can fill.
Additionally, the experiments seem to only look at two aspects: alignment of weights in matrix factorization (which is already known under weight decay, afaik), and bias toward less sharp minima (again, known). I would have liked to see a more direct test of the mechanism the paper claims is driving this movement toward flatter minima. At least, the balancing of the weights or some other direct result from the paper would have been desirable.

---

> ### Author Rebuttal · Authors · 2024-08-04
>
> Thanks for your review. Please first see our summary rebuttal above. We will answer concisely due to space constraint. However, please ask us to elaborate on every point that you need more detail.
>
> **Ref [1]...**
>
> Ref. [1] shows the existence of an invariant set, mainly due to the permutation symmetry. Our work focuses on continuous symmetry. We will include a reference to clarify this.
>
> **Ref [2] ...**
>
> This is a misunderstanding of our result. Our result shows that SGD can bias the solutions towards both sharper and flatter solutions, and this depends strongly on the data distribution. See our answers below on this point and the new experiment in Figure 7. We will add this reference and discussion to the manuscript.
>
> **Questions:**
>
> **Most derivations hold for symmetric Lie algebra elements...**
>
> This is due to your misreading. In the starting Definition 4.1, the exponential symmetry is explicitly defined with respect to a symmetric A. The reason is exactly what you point out -- if A is antisymmetric, the existence of Noether charge is questionable. We will add a note to clarify this below Def 4.1.
>
> Also,  we point out that antisymmetric matrices can have noether charge -- they are currently very poorly understood and hidden from simple analysis. As a counterexample, consider a rotation symmetry for (u,w) in 2d: the symmetry implies the following dynamics for GD: $wdu - udw = 0$. Simplifying the relation leads to (ignoring the sign problem for now) d(log(u) - log(w)) = 0. We have thus constructed a Noether charge for an antisymmetric-matrix symmetry: log(u) - log(w).
>
> **line 49: ref [40] wrong...**
>
> We will fix this.
>
> **Existence of C**
>
> Here, a technical remark on the antiderivative C is that one needs to assume that J is conservative, which is well known and so we omitted its discussion. We will add a note on this in the revision.
>
> **line 78: eq 5:... a. What is your argument for this G being the correct G?**
>
> Yes. You are correct.
>
> For point a, this is a good point that needs to be clarified. We made the specific choice that $G=\sqrt{\Sigma}$ in Eq. (1), but the goal was only to avoid introducing unnecessary technical details. Generally, one should use a general $G$ in the place of $\sqrt{\Sigma}$ such that $\Sigma = GG^T$, and, had we made this choice in Eq (1), what you suggested in point b will be the proof and so the result is the same. We will clarify this point.
>
> **"I don't agree with your statement..."**
>
> Thanks for this very deep question and suggestion (also, see the answer to point a). What you suggested is a more technical (and more correct) way of proving the relation, but there are multiple routes to get to Eq 31, and our original proof is correct. We also used a hidden condition that is rather obvious: the eigenvalues of $\sqrt{\Sigma}$ are the square roots of the corresponding eigenvalues of $\Sigma$. Thus, a zero eigenvalue of $\Sigma$ must correspond to a zero eigenvalue of $\sqrt{\Sigma}$. We will add a note on this.
>
> **line 83...**
>
> Our statement in line 83 does not contradict this statement. A point is still a "connected manifold" but of dimension zero.
>
> **line 104...**
>
> We will fix it.
>
> **eq 6...**
>
> A is defined to be symmetric.
>
> **line 114: why weight decay...**
>
> Thanks for this question. Here, the (fundamental) question is not "why" we include weight decay, but "what is the most general setting for which Theorem 4.3 holds correct?" It turns out that Theorem 4.3 holds true for any nonnegative $\gamma$, and this is the main reason why Def 4.2 allows for a weight decay.
>
> **Eq 7: the cross term...**
>
> We are not sure if we understand this question. There is no contribution from the weight decay term to $\Sigma$ because $\Sigma$ is a covariance of the gradient, whereas the weight decay only affects the expectation of the gradient, not its variance.
>
> **line 118...**
>
> By definition, A is symmetric. See our answer above.
>
> **line 214-215:**
>
> Thanks for this question. This example assumes the standard MSE loss for matrix factorization and is misplaced. We will move it below Eq. (13), and include a derivation of it.
>
> **eq 13: When $\gamma \neq 0$...**
>
> Yes, but this is not a problem. Because (please first see our reply above) Eq 13 can be decomposed into a sum of a symmetry term and weight decay (which does not have the A symmetry). So, this loss function obeys Def 4.2 and so Theorem 4.3 is applicable.
>
> **line 235...**
>
> These are not functions of the learning algorithm -- they are simply properties of the data points.
>
> **eq 39,40...**
>
> Matrix A is by definition symmetric.
>
> **line 531...**
>
> A is by definition symmetric.
>
>
> **Limitations:**
>
> **...alignment of weights in matrix factorization is already known under weight decay**
>
> We are afraid to point out that you have misunderstood our result, and our insight is not known.
> There are two aspects of alignment:
> 1. Alignment in norm: under SGD, the norm ratio is determined by the input and label covariance -- and so the balance in norm in general will NOT happen (even with weight decay), and this is a crucial new message we offer. Please examine Eq (14): only for very special $\Sigma_x$ and $\Sigma_\epsilon$, a balance in norm will happen.
> 2. Alignment in eigenvectors. Our crucial result is again that using SGD, the alignment will not occur even if weight decay is used.
>
> See Figure 7 in the pdf for additional experiments.
>
> **"and bias toward less sharp minima (again, known). "**
>
> This is also a misunderstanding. See the summary rebuttal and Figure 7-right for an experiment.
>
> **I would have liked to see a more direct test of the mechanism the paper...**
>
> Besides the result in Figure 7, another new insight is precise balance in the norm only happens in networks with at least two hidden layers, and, for a two-hidden layer network, only the two intermediate layers will always have balanced norms -- See Figure 8.

---

> > ### Comment · Reviewer_rhMv · 2024-08-12
> >
> > thank you for the answers. The point you make about Noether charge for rotations is interesting (is that mentioned elsewhere?). The log doesn't generalize easily to nonsquare matrices, but I think a variant of what you say may hold for higher dims. I Adlai see your did mention symmetric A in your definition. My general advice would be not to coopt general terms such as exponential symmetry for a subset of actual exponential symmetries. This is bad practice and misleads subsequent works that cite yours. So I suggest you use more specific expressions for Defs 4.1 and 4.2.
> > About Ref [2], yes I understand your result predicts bit ways depending on parameters. But the more useful fixed point is similar to ref 2. I think y should mention that work and then emphasize that your results can go both ways.
> > I am happy with your response and I think this is a significant paper. Therefore I am reading my score to 7.

---

> > > ### Author Response · Authors · 2024-08-13
> > > **reply**
> > >
> > > Thanks for your reply. We will revise our manuscript to make sure that the terminology is as accurate as possible.
> > >
> > > The existence of antisymemtric noether charge, to the best of our knowledge, has not been pointed out in any previous literature, and may be an important and interesting future problem to study.

---

> ### Author Response · Authors · 2024-08-12
> **Thanks for the feedback**
>
> Hi! We noticed that you have not communicated with us during the discussion period. It would be really helpful to us when revising and improving our manuscript if we could hear more from you.  We would be happy to hear any additional thoughts or feedback you have on our work and on our previous reply!

---

### Author Rebuttal · Authors · 2024-08-04

# Summary Rebuttal


We thank all the reviewers for their constructive feedback, which we will rely on to further improve our manuscript. We are encouraged to see that both reviewers **rhMv** and **QY9h** rate our contribution as "excellent." We believe some questions and criticisms may arise from misunderstandings or misinterpretations, which we aim to clarify below and in the revised manuscript.

To address the questions and criticisms from reviewers, we will include the following changes and additions in the revision. See the attached pdf for the additional experimental results.

1. **Balanced and Unbalanced norms**. An additional experiment in Figure 7-left shows that the norm balance and imbalance can be controlled by controlling the data distribution, which is in agreement with our theory. This shows one experiment that directly validates our theory and clarifies a misunderstanding of **rhMv** (namely, our theory shows that SGD can lead to both balanced and unbalanced norms of training, not just balanced ones)
2. **Convergence to flatter and sharper solutions**. An additional experiment in Figure 7-right shows that sharpness can be controlled by controlling the data distribution, which is in agreement with our theory. This shows one experiment that directly validates our theory and clarifies another misunderstanding of **rhMv** (namely, our result shows that SGD can lead to both flatter and sharper solutions, not just sharper ones). Both points 1 and 2 are novel insights, and it also novel (and perhaps surprising) that they can be explained by a single theory
3. **Deep linear nets**. Two additional experiments in Figure 8 on deep linear nets trained on MNIST (**QY9h, rhMv**). This directly confirms a theoretical prediction we made: the intermediate layers of a deep linear net converge to the same norm, while the first and last layers will be different
4. **Robustness Across Hyperparameters**. Three sets of additional experiments in Figure 9 show the robustness of the prediction to different hyperparameter choices such as the data dimension, learning rate, and batch size (**HyRa**)
4. **Technical Clarifications**. Additional clarification of technical and mathematical details (**rhMv, QY9h, HyRa**)
5. **Experimental Detail**. Additional detail and clarification of experimental setting (**HyRa**)
6. **References**. Discussion of missed references (**rhMv**)

HyRa named three weaknesses that we believe to be due to misunderstanding:

1. **Insufficient mathematical preliminaries**: We believe this is due to potential misreading of our equations and unfamiliarity with related mathematical tools and literature within the ML community.
2. **Strong assumption**: This stems from a misunderstanding of the problem setting of section 5.1 -- as we make no assumption in section 5.1.
3. **Explanation of Contributions**: This may result from unfamiliarity with related fields such as deep learning theory, neuroscience, and stochastic differential equations. Our work is interdisciplinary, requiring a broader perspective beyond practical deep learning to appreciate its contributions fully.

We have addressed these points from HyRa in detail below.

We look forward to your feedback on these revisions and are open to further discussion to enhance the manuscript.

---

### Decision · Program_Chairs · 2024-09-25

**Decision:**

Accept (poster)

**Comment:**

The paper studied the learning dynamics of SGD when there is a symmetry in the loss function (called exponential symmetry).
Reviewer rhMv and QY9h appreciate the contributions of the paper. Reviewer HyRa offered a bolderline review, but agreed upon positive ratings.

Overall, I recommend accept.